# Seasonal variation of the principal tidal constituents in the Bohai Sea

Daosheng Wang[1,2,3], Haidong Pan[4], Guangzhen Jin[5], Xianqing Lv[4]

[1] College of Marine Science and Technology, China University of Geosciences, Wuhan 430074, China
[2] Southern Marine Science and Engineering Guangdong Laboratory (Guangzhou), Guangzhou 511458, China
5    [3] Shenzhen Research Institute, China University of Geosciences, Shenzhen 518057, China
[4] Physical Oceanography Laboratory/CIMST, Ocean University of China and Qingdao National Laboratory for Marine Science and Technology, Qingdao 266100, China
[5] School of Marine Sciences, Sun Yat-Sen University and Key Laboratory of Marine Resources and Coastal Engineering in Guangdong Province, Guangzhou 510275, China

10    *Correspondence to*: Xianqing Lv (xqinglv@ouc.edu.cn)

**Abstract.** The seasonal variation of tides plays a significant role in water level changes in coastal regions. In this study, seasonal variations of four principal tidal constituents, $M_2$, $S_2$, $K_1$, and $O_1$, in the Bohai Sea, China, were studied by applying an enhanced harmonic analysis method to two time series: one-year sea level observations at a mooring station (named E2) located in the western Bohai Sea and 17-year sea level observations at Dalian. At E2, the $M_2$ amplitude and phase lag have annual frequencies, with large values in summer and small values in winter, while the frequencies of $S_2$ and $K_1$ amplitudes are also nearly annual. In contrast, the $O_1$ amplitude increases constantly from winter to autumn. The maxima of phase lags appear twice in one year for $S_2$, $K_1$ and $O_1$, taking place near winter and summer. The seasonal variation trends estimated by the enhanced harmonic analysis at Dalian are different from those at E2, except for the $M_2$ phase lag. The $M_2$ and $S_2$ amplitudes show semi-annual and annual cycles, respectively, which are relatively significant at Dalian. The results of numerical experiments indicate that the seasonality of vertical eddy viscosity induces seasonal variations of the principal tidal constituents at E2. However, the tested mechanisms, including seasonally varying stratification, vertical eddy viscosity and mean sea level, do not adequately explain the observed seasonal variations of tidal constituents at Dalian.

## 1 Introduction

Tidal motion is one of the major dynamical processes in the Bohai Sea and has been widely studied (Fang et al., 2004).
25   Although there is no primary seasonal cycle in the moon's orbit, a significant seasonal variation in the principal lunar tidal constituent has been observed and is dominant in coastal and polar regions (Müller et al., 2014). The seasonal variations of several semidiurnal tides are also found to be significant in the Bohai Sea (Fang et al., 1986), but the corresponding comprehensive investigations are sparse, which results in that the seasonal variation characteristics and mechanisms need to be further studied.

30   The seasonal variation in the major tidal constituent $M_2$ has received considerable attention (Gräwe et al., 2014). Corkan (1934) inferred a seasonal modulation of the $M_2$ tide by analysing several sea level records near the British coast. Foreman et

al. (1995) observed a seasonal cycle of the $M_2$ amplitude at Victoria, which is on the southern tip of Vancouver Island off Canada's Pacific coast. Kang et al. (1995) revealed the seasonal variability of the $M_2$ harmonic constants in the seas adjacent to Korea. Huess and Andersen (2001) found a seasonal variation in the $M_2$ constituent in the northwest European shelf. Kang et al. (2002) investigated the seasonal variability of the $M_2$ tide in the Yellow and East China Seas. Georgas (2012) observed seasonal episodes of significant tidal damping and modulation in the Hudson River estuary. Müller et al. (2014) studied the global seasonal cycle of the $M_2$ tide and found significant seasonal variations in several coastal areas, including the North Sea, East China Sea and Yellow Sea, Sea of Okhotsk and regions of the Banda, Timor, and Arafura Seas north of Australia. Tazkia et al. (2017) found that the $M_2$ amplitude changed markedly between winter and summer in the northern Bay of Bengal. Several other studies have analysed the seasonal variability of the $M_2$ tide in polar regions. Mofjeld (1986) observed seasonal fluctuations of the tidal harmonic parameters on the north-eastern Bering Sea shelf. Kagan and Sofina (2010) showed that the seasonal variability of tidal constituents was widespread in the Arctic Ocean. Further, Müller et al. (2014) studied the global seasonal cycle of the $M_2$ tide and also identified significant seasonal variations in the Arctic regions.

Most previous studies primarily focus on the seasonal variation in the $M_2$ amplitude without considering the seasonality of other tidal constituents and their phase lags (Gräwe et al., 2014). However, several studies have investigated the seasonality of multiple constituents. For example, Fang and Wang (1986) studied the seasonal variations of $M_2$, $N_2$, $O_1$ and $M_4$ in the Bohai Sea by introducing astro-meteorological constituents; Devlin et al. (2018) found that the diurnal ($K_1$ and $O_1$) and semidiurnal ($M_2$ and $S_2$) amplitudes and phase lags exhibited strong seasonal variability in the seas of Southeast Asia.

In this study, sea level observations at one mooring station (E2) and one tidal gauge station (Dalian) in the Bohai Sea were used to investigate the seasonal variability of the principal tidal constituents $M_2$, $S_2$, $K_1$ and $O_1$ with the enhanced harmonic analysis (EHA). The rest of the paper is organised as follows. In Section 2, the sea level observations in the Bohai Sea are reported and the analysis methods are described. In Section 3, the seasonal variability of the principal tidal constituents is estimated by analysing observations. The mechanisms underlying the seasonal variability are discussed by using numerical experiments in Section 4. Further discussions and conclusions occupy Sections 5 and 6, respectively.

## 2 Observations and methods

### 2.1 Observations

From 0000 UTC 1 November 2013 to 0000 UTC 1 November 2014, total sea levels were observed hourly using a moored pressure gauge accurate to within 5 cm (Lv et al., 2019), at E2 station (38.65°N, 118.28°E) in the Bohai Bay, China (Figure 1). The time series of the total sea levels at E2 is shown in Figure 2a, demonstrating the continuous coverage of the observations. The obvious sea level drops in November 2013 at E2 are mainly caused by the winter storms.

Hourly sea level data at the Dalian tidal gauge station used in this study were obtained from the University of Hawaii Sea Level Center. As indicated in Feng et al. (2015), the stations at Dalian and Laohutan (7 km away) have data for the period 1975-1990 and 1991-1997, respectively; in addition, the Dalian station was relocated in 1976, and again in 1979, after which

Dalian station shared the same position as Laohutan. Therefore, following Feng et al. (2015), the tidal gauge station data at Dalian used in this study were comprised of data from Dalian from 1980-1990 and that from Laohutan from 1991-1997, as shown in Figure 2b.

## 2.2 Classical harmonic analysis

A sea level is composed of components from different sources (Godin, 1972; Foreman, 1977; Fang et al., 1986; Pawlowicz et al., 2002; Foreman et al., 2009):

$$\zeta(t) = \zeta_0 + \sum_{k=1}^{K} \left\{ f_k(t) A_k \cos\left[V_k(t) + u_k(t) - g_k\right] \right\} + R(t)$$
$$= \zeta_0 + \sum_{i=1}^{N_{NR}} \left\{ f_i(t) A_i \cos\left[V_i(t) + u_i(t) - g_i\right] \right\} + R(t) + \tag{1}$$
$$\sum_{j=1}^{N_R} \left\{ f_j(t) A_j \cos\left[V_j(t) + u_j(t) - g_j\right] + \sum_{n=1}^{N_I} f_n(t) A_n \cos\left[V_n(t) + u_n(t) - g_n\right] \right\}$$

where $\zeta(t)$ is the total sea level; $\zeta_0$ is the mean sea level; $A$ and $g$ are the amplitude and phase lag (UTC time, the same below), respectively; $f$ and $u$ are the nodal corrections to amplitude and phase lag, respectively; $V$ is the astronomical argument;

$R$ is the nontidal component; $K$ is the number of tidal constituents; $N_{NR}$ is the number of non-reference constituents; $N_R$ is the number of reference constituents; and $N_I$ is the number of constituents to be inferred from the $j^{th}$ reference constituent.

The mean sea level, amplitude and phase lag of each constituent can be solved by analyzing a time series of sea level observations at a specific point using classical harmonic analysis (CHA). With different assumptions and conditions, CHA can be performed using the T_TIDE (Pawlowicz et al., 2002), U_TIDE (Codiga, 2011) or Institute of Ocean Sciences Tidal

Package (Foreman et al., 2009). In this study, T_TIDE, in which the astronomical argument varies linearly and the nodal correction is performed after least squares fit, is used to realize CHA.

## 2.3 Segmented harmonic analysis

Following Foreman et al. (1995), Kang et al. (1995), Müller et al. (2014) and Devlin et al. (2018), sea level observations are divided into monthly segments by calendar month and CHA with nodal and inference corrections is applied to each monthly

segment to obtain the discrete tidal harmonic parameters (i.e., amplitude and phase lag). Then the discrete amplitude and phase lag for each month are interpolated using cubic spline interpolation to obtain the temporally varying amplitudes and phase lags. This methodology is termed segmented harmonic analysis (SHA). Following Kang et al. (1995), one monthly segment is analyzed only when the duration of the observations is longer than 26 days.

## 2.4 Enhanced harmonic analysis

By combining CHA with independent point scheme and cubic spline interpolation, Jin et al. (2018) developed EHA to directly obtain temporally varying mean sea level and tidal harmonic parameters. In contrast, the harmonic parameters are

assumed to be constant in CHA and constant within each month in SHA. A MATLAB toolkit, S_TIDE, was released to realize EHA by Pan et al. (2018b). In this study, nodal and astronomical argument corrections are embedded into the least square fit, following Foreman et al. (2009); in addition, the harmonic parameters of the minor tidal constituents are assumed to be constant and calculated together with the temporally varying harmonic parameters of the principal tidal constituents to resolve more constituents and retain computational stability. The sea level in EHA is as follows:

$$\zeta(t) = \zeta_0(t) + \sum_{i=1}^{I}\left\{ f_i(t)A_i(t)\cos\left[V_i(t) + u_i(t) - g_i(t)\right]\right\} + \sum_{j=1}^{J}\left\{ f_j(t)A_j\cos\left[V_j(t) + u_j(t) - g_j\right]\right\} \tag{2}$$

where $I$ is the number of principal tidal constituents with temporally varying harmonic parameters; $J$ is the number of minor tidal constituents with constant harmonic parameters; and the mean sea level and nontidal component are included in $\zeta_0(t)$.

Similar to Jin et al. (2018) and Pan et al. (2018b), the independent point scheme and cubic spline interpolation are used to jointly solve the temporally varying and constant harmonic parameters, which are not shown here for brevity. As mentioned in Pan et al. (2018b), the temporally varying harmonic parameters obtained using EHA with different numbers of independent points represent fluctuations on different time scales. In this study, six independent points are used to obtain the seasonal variability of the principal tidal constituents.

## 3 Results

One-year sea level observations at E2 were analyzed using CHA with the automated constituent selection algorithm (Pawlowicz et al., 2002). According to the signal-to-noise ratio (Pawlowicz et al., 2002; Matte et al., 2013), $M_2$, $K_1$, $S_2$ and $O_1$ were selected as the principal tidal constituents to be investigated in this study.

### 3.1 Seasonal variability at E2

As shown in Figure 3, the significant constituent near $K_1$ was $P_1$, which was unable to be resolved when analysing one-month observations (Fang and Wang, 1986), while that for $S_2$ was $K_2$ in the semidiurnal frequency band. Therefore, when the monthly analysis was performed in SHA, the automated constituent selection algorithm in T_TIDE was used to determine the analysed constituents; in addition, the unresolved constituents $P_1$ and $K_2$ were inferred from $K_1$ and $S_2$, respectively, with the inference parameters taken from a yearly harmonic analysis of the one-year sea level observations at E2 (Kang et al., 1995; Foreman et al., 2009). When EHA was used to directly analyse the sea level observations at E2, the harmonic parameters of $M_2$, $K_1$, $S_2$, $O_1$, $P_1$ and $K_2$ were estimated together, among which the harmonic parameters of $M_2$, $K_1$, $S_2$ and $O_1$ were assumed to be temporally varying and those of $P_1$ and $K_2$ were assumed to be constant.

As shown in Figure 4, the estimated harmonic parameters obtained with SHA and EHA, including the temporally varying amplitudes and phase lags, were nearly equal and the averaged values were near to that estimated using CHA, indicating that the temporal variations in the harmonic parameters of the principal tidal constituents at E2 can be reasonably estimated using both SHA and EHA. Based on Wei and Wang (2012) and Zhang et al. (2017), spring, summer, autumn and winter were defined

as March to May, June to August, September to November and December to February of the following year, respectively. The temporally varying harmonic parameters of the principal tidal constituents showed seasonal variations (Figure 4). For $M_2$, the seasonal variations were significant: both amplitude and phase lag reached maximum in summer and minimum in winter, as in Müller et al. (2014). The seasonality of the $S_2$ amplitude was not significant, but the estimated results using EHA increased

significantly in summer. The temporal variation of the $K_1$ amplitude spanned one year, with maxima and minima in summer and winter, respectively. In stark contrast, the $O_1$ amplitude increased from winter to autumn. The phase lags of the $S_2$, $K_1$ and $O_1$ components showed semi-annual cycles: larger in summer and winter and smaller in spring and winter, respectively. Among the four principal tidal constituents, only the $M_2$ amplitude had the similar variation trend as the phase lag.

        The seasonally averaged amplitudes and phase lags of the principal tidal constituents are listed in Table 1. The variation

trends of the averaged harmonic parameters of these constituents were the same as those in Figure 4. Compared to the annual averages, the seasonal mean $M_2$ amplitude increased by 6.90 cm (approximately 9.33%) in the summer and decreased by 6.68 cm (approximately 9.03%) in the winter, close to the estimated values in Foreman et al. (1995) (6%), Huess and Andersen (2001) (6%) and Müller et al. (2014) (5%–10%). For $S_2$ ($K_1$), the seasonally averaged amplitudes decreased by 4.71% (7.72%) in the winter and increased by 7.93% (5.91%) in the summer, indicating a nearly annual cycle as shown in Figure 4. The

seasonal mean $O_1$ amplitude in the summer increased by 3.45 cm compared to that in the winter. The $M_2$ phase lag in winter was smaller than its annual average, and all the other three principal tidal constituents shared a different pattern: values in both winter and summer were larger than the corresponding annual average.

### 3.2 Seasonal variability at Dalian

        The multiyear data at Dalian shown in Figure 2b were analysed year by year. In each year, one-year sea level observations

were analysed using CHA, SHA and EHA with similar settings to E2. As shown in Figure 5, $P_1$ and $K_2$ were the significant constituents unresolved in the monthly analysis, just like at E2. Therefore, $P_1$ and $K_2$ were inferred from $K_1$ and $S_2$ in SHA and taken as minor constituents with constant harmonic parameters in EHA. The estimated harmonic parameters from various years were then averaged (Fang and Wang, 1986) and are shown in Figure 6. The averaged harmonic parameters estimated using both the SHA and EHA were near to those obtained using CHA, showing that the estimated results were reasonable. In

addition, the estimated harmonic parameters obtained using EHA were much closer to those obtained using SHA for data at Dalian than those at E2.

        The variation trends of the harmonic parameters estimated using EHA at Dalian were different from those at E2, except for the $M_2$ phase lag (Figure 6). The $M_2$ amplitude at Dalian showed a semi-annual cycle, with large values in summer and winter and small values in spring and autumn, respectively. The $S_2$ amplitude had significant annual cycle, with maximum in

winter and minimum in summer, which is opposite of the variation trend of $S_2$ amplitude at E2. The $K_1$ amplitude was nearly constant from winter to spring and increased during the summer. The $O_1$ amplitude reached the minimum in the winter and summer while increasing slightly in the spring and autumn. The estimated $S_2$ phase lag reached the maximum in the spring

with small variation. The $K_1$ and $O_1$ phase lags had the same trend, increasing in the winter and summer while decreasing in the spring and early autumn.

The averaged amplitudes and phase lags of the principal tidal constituents at Dalian, as listed in Table 2, showed a seasonal variation that was generally smaller than that at E2. All of the seasonal changes of the principal tidal constituents were less than 1.80 cm, which is the case only for $S_2$ amplitude at E2. In addition, all of the seasonal changes of the phase lags were less than 2.20°, while the $S_2$ and $K_1$ phase lags at E2 changed by at least 5.00° in the summer and winter, respectively. The relative change of the $M_2$ amplitude at Dalian was less than 1%, which was significantly less than that at E2. The relative changes of all phase lags were less than 1% except for the $M_2$ tide which was larger than 2% in both summer and winter compared to the annual averages and larger than those at E2. An increase in the $S_2$ amplitude of 5.34% occurred in the winter, larger than its decrease in the winter at E2.

In summary, the harmonic parameters of the principal tidal constituents at E2 and Dalian varied seasonally but with different patterns. The amplitude of the principal tidal constituent $M_2$ at E2 showed an annual cycle, while that at Dalian had a semi-annual cycle. The $M_2$ phase lags at E2 and Dalian had the similar variation trend, with lager values in summer and small values in winter. The $S_2$ amplitude in winter at E2 was less than that in summer, which was opposite to that at Dalian. The $K_1$ amplitude at E2 had an annual frequency, with large values in summer and small values in winter, while the $O_1$ amplitude increased steadily. In contrast, the variations of the $K_1$ and $O_1$ amplitudes at Dalian were small. The maxima of the $S_2$, $K_1$ and $O_1$ phase lags at E2 appeared twice a year, like those of $K_1$ and $O_1$ and different from that of $S_2$ at Dalian.

## 4 Mechanisms for the seasonal variability

Several previous studies have investigated the seasonal variability of the $M_2$ amplitude. Three main mechanisms have been proposed:

1) Seasonal variations of the mean sea level. Corkan (1934) related the seasonal modulation of the $M_2$ tide near the British coast to seasonal variations of sea level and atmospheric pressure. Tazkia et al. (2017) pointed out that the seasonal variability of the sea level generated by many processes can induce a seasonal variation of the $M_2$ tide, as tidal wave propagation was controlled by water depth on the first order.

2) Seasonally varying stratification. Foreman et al. (1995) presumed that the seasonal variability of the $M_2$ amplitude at Victoria, Canada was induced by the changes in stratification due to seasonal variability in estuarine flow. Kang et al. (2002) used a two-layer numerical model to investigate the baroclinic response of the tide and tidal currents in the Yellow and East China Seas, and found that seasonal stratification had several noticeable effects on the tides, including varying degrees of current shear, frictional dissipation, and barotropic energy flux. Müller (2012) indicated that in shallow seas, seasonal variations in stratification were a major factor for the observed seasonal modulation in tides. Müller et al. (2014) pointed out that the seasonal changes in stratification on the continental shelf affected the vertical profile of the eddy viscosity to further cause the seasonal variability of the $M_2$ tide.

3) Seasonally varying ice coverage. St-Laurent et al. (2008) proposed that the significant seasonal variations of the $M_2$ surface elevation in all regions of the Hudson Bay system were essentially caused by under-ice friction. Georgas (2012) pointed out that the seasonal episodes of significant tidal damping (reductions in tidal amplitudes by as much as 50%) observed in the Hudson River estuary were primarily caused by the under-ice friction as well. Müller et al. (2014) found that the frictional effect between the sea ice and ocean surface layer led to the seasonal variability of the $M_2$ tide. The Bohai Sea in north China freezes to varying degrees every winter for approximately 3–4 months (Su and Wang, 2012). According to the back-effect connection of the coastal shelf and open ocean via resonance mechanisms (Arbic et al., 2009; Arbic and Garrett, 2010), sea ice may be important to the seasonality of principal tidal constituents. However, Zhang et al. (2019) performed numerical experiments with a three-dimensional ice-ocean coupled model and found that the damping effect of sea ice on the astronomical tides was almost negligible in the Bohai Sea. Therefore, ice coverage was not considered in this study.

Other mechanisms, including long-term changes in the tidal potential (Molinas and Yang, 1986), monsoon (Devlin et al., 2018), interactions with other physical phenomena (Huess and Andersen, 2001; Pan et al., 2018a), changes in the internal tide with corresponding small changes in its surface expression (Ray and Mitchum, 1997; Colosi and Munk, 2006), as well as a number of technical reasons, may also change the $M_2$ amplitude on various time scales. The above reasons have been presented or discussed in Woodworth (2010), Müller (2012), Müller et al. (2014), Tazkia et al. (2017), and Talke and Jay (2020).

## 4.1 Design of numerical experiments

Several numerical experiments (Exp1–Exp4) were carried out to simulate the four principal tidal constituents in the Bohai Sea under different conditions using MITgcm (Marshall et al., 1997), testing the influence of seasonal variations of mean sea level and stratification on the seasonal variability of the principal tidal constituents.

Identical model settings used in all of the numerical experiments were described as follows. The simulation area of the Bohai Sea is shown in Figure 1b. The horizontal resolution was $2'\times2'$ and there were 16 layers in the vertical direction with thicknesses ranging from 2-5 m. The four principal tidal constituents $M_2$, $S_2$, $K_1$, and $O_1$ were implemented as tidal forcing at the east open boundary, whose data were predicted using the constant harmonic parameters extracted from the TPXO model (Egbert and Erofeeva, 2002). Sea surface boundary conditions were not considered. The horizontal eddy viscosity coefficient was set to $1.0\times10^3$ m$^2$/s, and the quadratic bottom drag coefficient was set to $1.3\times10^{-3}$ (Wang et al., 2014). The integral time step was 60 s and the total simulation time was 60 d. The results of the final 30 d were used to calculate the harmonic parameters using CHA with the automated constituent selection algorithm.

Details of the model settings for numerical experiments Exp1–Exp4 are listed in Table 3. In Exp1, the simulation started from 0000 UTC 1 January 2014, while the simulation started from 0000 UTC 1 July 2014 in Exp2-Exp4. In Exp1, horizontally homogeneous profiles of the initial temperature and salinity (Figure 7) were extracted from the HYCOM global analysis results in winter, while those in summer were used in Exp2–Exp4. The vertical eddy viscosity coefficient was specified directly and no turbulence closure schemes were used. In Exp1, the vertical eddy viscosity coefficient was set to $2.0\times10^{-3}$ m$^2$/s through a trial and error procedure. According to Müller et al. (2014), the eddy viscosity in summer was reduced by orders of magnitude

compared to well-mixed conditions in winter, as the stratification stabilized the water column. Therefore, the vertical eddy viscosity coefficient was decreased by one-half in Exp3 to test the influence of the vertical eddy viscosity caused by the stratification. As shown in Figure 8, monthly means of the low-pass sea levels, filtered using a cosine-Lanczos filter with a high frequency cut-off of 0.8 cpd, were nearly equal to the estimated mean sea level using SHA. They exhibited the same variation trend as those obtained using EHA, with large values in summer and small values in winter. As the difference between the averaged mean sea level in summer and that in winter was about 0.2 m, Exp4 included 0.2-m increase of water depth to test the influence of mean sea level.

## 4.2 Modelling Results

The simulated harmonic parameters of the four principal tidal constituents in the numerical experiments and those obtained from observations at E2 and Dalian are shown in Figure 9. The simulated harmonic parameters were a little far from the observed results, except the $M_2$ amplitude at E2 simulated in Exp1 and that simulated in Exp2, possibly because the constant bottom drag coefficient was used (Wang et al., 2014) and the ocean circulation and other important factors were not considered. However, the differences between the simulated results in the different numerical experiments can be used to display the influence of potential factors on the seasonal variability of the principal tidal constituents.

The observed amplitudes at E2 in summer were larger than those in winter for all four principal tidal constituents, as shown in Figure 9. However, the simulated amplitudes in Exp2 were nearly equal to those in Exp1. In contrast, both the decreased vertical eddy viscosity coefficient in Exp3 and the increased mean sea level in Exp4 increased the amplitudes for all principal tidal constituents. The increases of the observed $M_2$ and $S_2$ amplitudes at E2 from winter to summer were 13.58 cm and 2.62 cm, respectively, while those were 13.34 cm (2.08 cm) and 2.75 cm (0.56 cm) for simulated results in Exp3 (Exp4) compared to those in Exp1. In addition, the increases of the observed $K_1$ and $O_1$ amplitudes were also captured better by the simulated results in Exp3 than those in Exp4, as shown in Figure 9. Therefore, the seasonally varying amplitudes of all principal tidal constituents were primarily caused by the seasonal variation of vertical eddy viscosity. For $M_2$, $S_2$ and $O_1$ tides, the variation trend of simulated phase lags between Exp3 and Exp1 shared the same pattern with the observed variations between summer and winter, indicating the effects of the seasonally varying vertical eddy viscosity. In contrast, Exp2 with the changes in stratification and Exp4 with changes in mean sea level only reproduced the variation trend of the $S_2$ and $O_1$ phase lags, respectively. The aforementioned results demonstrated that seasonal variation in the vertical eddy viscosity was the most important mechanism influencing the seasonal variability of principal tidal constituents at E2.

The observed $S_2$ amplitude at Dalian was larger in winter than in summer. The simulated result in Exp2 (summer) showed a decrease from Exp1 (winter) while those in Exp3 and Exp4 were larger than that in Exp1, indicating the seasonality of stratification as a possible reason. However, the simulated seasonal variation between Exp2 and Exp1 was too weak, which was less than 1 cm, possibly because the simple horizontally homogeneous temperature and salinity profiles could not reflect reality. The water depth is large in the eastern part of Bohai Sea (Figure 1b), so the stratification and ocean circulation were noteworthy and had significant effects on the tides. The increases of the $M_2$ and $O_1$ amplitudes were only captured by Exp3

with changes in the vertical eddy viscosity coefficient and Exp2 with changes in stratification, respectively. The variation trends of the $M_2$ and $S_2$ phase lags were not reproduced well in any experiments, among which Exp2 performed the best, while those of $K_1$ and $O_1$ were best captured by Exp3, where the simulated results were smaller than those in Exp1. On the whole, the seasonal variations of the principal tidal constituents at Dalian were not adequately reproduced by the numerical experiments, indicating that all the tested mechanisms were not the possible mechanism.

The variations of the simulated amplitudes from winter (Exp1) to summer (Exp3) in the entire Bohai Sea are shown in Figure 10. The spatial distribution of the variations in $M_2$ amplitude had a strong positive correlation (R=0.98) with that in the $S_2$ amplitude, similar to that for the diurnal tides (R=0.98). Furthermore, the distributions were possibly related to tidal wave propagation as their patterns were similar to the co-phase lines, as shown in Figure 10. For the semi-diurnal tides $M_2$ and $S_2$, the simulated amplitudes in summer were larger than those in winter in Bohai Bay, Laizhou Bay, and Liaodong Bay, which was the same as that obtained by analysing the sea level data at several tidal gauge stations in the Bohai Sea in Fang and Wang (1986) and that simulated by numerical model in Kang et al. (2002), while the simulated results in summer were smaller than those in winter in the middle of the Bohai Sea. The spatial distribution of the absolute differences between the $M_2$ amplitude in summer and that in winter was similar to that in Müller et al. (2014). For the diurnal tides $K_1$ and $O_1$, the simulated amplitudes in summer were larger than those in winter in Bohai Bay, Laizhou Bay, Liaodong Bay and the middle areas, while smaller in the northeast part of the Bohai Strait.

## 5 Discussions

In this study, the EHA developed in Jin et al. (2018) and Pan et al. (2018b) was further improved in order to resolve more tidal constituents by adding the minor constituents whose harmonic parameters were assumed to be constant and calculated together with the temporally varying harmonic parameters of the principal tidal constituents. The nodal and astronomical argument corrections were embedded into the least square fit to eliminate the influences of nodal cycle and linearly varying astronomical argument. In fact, there have been multiple improvements to T_TIDE in the past decades, such as R_T_TIDE (Leffler and Jay, 2009), versatile tidal analysis (Foreman et al., 2009), U_TIDE (Codiga, 2011) and NS_TIDE (Matte et al., 2013). In R_T_TIDE, versatile tidal analysis and U_TIDE, the harmonic parameters (i.e., amplitude and phase lag) are assumed to be constant. However, harmonic parameters are not constant and have multiscale temporal variations, as shown in researches such as Corkan (1934), Kang et al. (1995), Müller et al. (2014), Devlin et al. (2018), and Talke and Jay (2020). Neglecting seasonal variation of tides will introduce significant error in sea level prediction (Fang and Wang, 1986). EHA assumes the harmonic parameters of the principal tidal constituents are temporally varying and incorporate their calculation into the least squares fit, which is an important improvement to T_TIDE. In NS_TIDE, the harmonic parameters are also assumed to be temporally varying. However, the harmonic parameters are taken as functions of river flow and greater diurnal tidal range at the reference station. Therefore NS_TIDE can only be applied to river tides, while EHA can be applied in analyzing any time series. On the whole, EHA used in this study is indeed superior to other methods.

It is noted that the duration of sea level observations at E2 is from 2013 to 2014 and that for Dalian is from 1980 to 1997, which are from different eras and may defy comparison because the 1997-1998 El Nino is one of the strongest in the twentieth century (Chavez et al., 1999) and changes many of the physical oceanic properties (Nezlin and Mcwilliams, 2003; Shang et al., 2005; Liu et al., 2010). In addition, the duration of hourly sea level observations at E2 used in this study is only one year, which is also a limitation. Although the different time eras and the rare event might slightly skew seasonal pattern of tides, the seasonal variations of the principal tidal constituents $M_2$, $S_2$, $K_1$ and $O_1$ obtained using EHA are the same as those using traditional SHA, indicating that the results are not unreasonable and can reflect the seasonal variations of tides in the analysis period. It is a limitation that only the sea level observations at E2 and Dalian are analyzed, so the studies using much more non-publicly available data to further investigate the seasonality of tides in the Bohai Sea are encouraged. The strong seasonal variation trends of the principal tidal constituents at E2 can be captured by the results of numerical experiments. The multi-annually averaged results at Dalian also showed the seasonal variations of the principal tidal constituents, but the results of numerical experiments were not in good accordance with the observed results, which may be because horizontally homogeneous profiles of the initial temperature and salinity were used and the temporally varying ocean circulation was not considered. In addition, the constant harmonic parameters were used to predict the sea level at the eastern open boundary that near the Dalian, which may be another reason for the disagreement between the observed and simulated seasonal variations of the tidal constituents.

The seasonality of the principal tidal constituents has been investigated widely. As shown in Müller et al. (2014), there were significant seasonal variations in $M_2$ tide in several coastal regions and the maximum annual tide was in July (±1 month) in most of the ocean. However, the spatial and temporal inhomogeneities also existed and the summer amplitudes of $M_2$ tide were less than those in winter in several areas, as shown in Kang et al. (1995), Müller et al. (2014), Devlin et al. (2018) and Figure 10 in this study. In general, the $M_2$ tidal amplitude in summer was larger than that in winter in many areas, such as the Bohai Sea (Fang and Wang, 1986), the North Sea (Huess and Andersen, 2001; Gräwe et al., 2014; Müller et al., 2014), most of the Ganges-Brahmaputra-Meghna delta (Tazkia et al., 2017), the seas of Southeast Asian (Devlin et al., 2018), Liverpool (Corkan, 1934), Victoria (Foreman et al., 1995), the western part of the Yellow and East China Seas (Kang et al., 2002), the Hudson Bay and Foxe Basin (St‑Laurent et al., 2008), which was the same as that obtained by analysing the observations at E2 and Dalian in this study. Devlin et al. (2018) found that the diurnal and semidiurnal tidal amplitudes and phases exhibited a high degree of seasonality in the seas of Southeast Asia, and Fang and Wang (1986) indicated that the $O_1$ amplitude in summer was also larger than that in winter in the Bohai Sea, which were similar to those concluded in this study. The seasonality of the principal tidal constituents obtained in the study was mainly similar to those in previous studies, but the novel EHA was firstly used to estimate the seasonal variations of the principal tidal constituents and the numerical experiments using three-dimensional MITgcm were performed to explore the physical mechanisms.

The seasonal variations of stratification and vertical eddy viscosity and their influences on the tidal amplitudes may be as follows. In winter, the strong northwest Asia monsoon develops a vertically well-mixed condition (Yanagi et al., 2001; Jeon et al., 2014). The vertically well-mixed condition will not stabilize the tidal currents and lose more energy, leading to smaller

tidal amplitudes. As the surface heating rate and freshwater discharge increase in summer, the mixing is insufficient to homogenize the input potential energy and cause stratified conditions (Huang et al., 1999; van Haren, 2000). Hence the reduced vertical eddy viscosity will increase the tidal amplitudes. However, the $S_2$ amplitude at Dalian was larger in winter and smaller in summer, which is inconsistent with the other principal tidal constituents. It may be related with the atmospheric variations and should be further investigated in future studies.

## 6 Conclusions

In this study, based on one-year sea level observations at E2 and 17-year sea level observations in the Bohai Sea, the seasonal variability of the principal tidal constituents was investigated using different methods. In analysis of sea level observations at E2 and Dalian, the seasonal variations of all principal tidal constituents obtained using EHA were nearly equal to those obtained using SHA (Figures 4 and 6), indicating that the seasonal variations were not related to the applied methods. At both E2 and Dalian, the principal tidal constituents $M_2$, $S_2$, $K_1$ and $O_1$ exhibited seasonal variations (Figures 4 and 6). The $M_2$ amplitude at E2 had an annual cycle, while that at Dalian showed a semi-annual cycle. The $M_2$ phase lags at E2 and Dalian had the similar variation trend, with large values in summer and small values in winter. The $S_2$ amplitude in winter at E2 was less than that in summer, which was opposite to that at Dalian. The $K_1$ amplitude at E2 had an annual cycle, with large values in summer and small values in winter, while the $O_1$ amplitude increased steadily. On the contrary, the variations of the $K_1$ and $O_1$ amplitudes at Dalian were small. The maxima of the $S_2$, $K_1$ and $O_1$ phase lags at E2 appeared twice a year, which was the same as those of $K_1$ and $O_1$ and different from that of $S_2$ at Dalian.

Through several numerical experiments, the mechanisms of the seasonal variability of the principal tidal constituents were investigated. Although the simulated harmonic parameters of four principal tidal constituents were not consistent well with the observations in most cases, the differences between the simulated results in summer and winter indicated that the seasonal variations of the principal tidal constituents at E2 were caused by the seasonality of the vertical eddy viscosity, while the seasonal variations at Dalian were not reproduced by the test mechanisms, including seasonally varying stratification, vertical eddy viscosity and mean sea level. Therefore, taking into consideration of temporally varying harmonic parameters, the synchronous simulation of circulation and tides and a reasonable parameterization scheme to convert the variations in stratification to those in vertical eddy viscosity were needed for precise simulation of the tides.

## Data availability

The HYCOM global analysis data is available at http://hycom.org. New version of S_TIDE package can be downloaded from https://www.researchgate.net/project/Adaptation-of-tidal-harmonic-analysis-to-nonstationary-tides. The hourly sea level observations at Dalian are available at https://uhslc.soest.hawaii.edu/datainfo/. The hourly sea level observations at E2 used in this work are available from the authors upon request (xqinglv@ouc.edu.cn).

# Acknowledgements

The authors would like to thank the editors and anonymous reviewers for their constructive suggestions to greatly improve the manuscript and Zheng Guo for polishing the manuscript.

This work was supported by the National Key Research and Development Program of China (Grant No. 2017YFC1404700), the Key Special Project for Introduced Talents Team of Southern Marine Science and Engineering Guangdong Laboratory (Guangzhou) (Grant No. GML2019ZD0604), the Discipline Layout Project for Basic Research of Shenzhen Science and Technology Innovation Committee (Grant No. 20170418), the Guangdong Special Fund Program for Economic Development (Marine Economic) (Grant No. GDME-2018E001) and the Three Big Constructions: Supercomputing Application Cultivation Projects.

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

**Table 1. Averaged amplitudes (cm) and phase lag (°) of the principal tidal constituents obtained using EHA at E2**

| Constituents | Parameter | Annual | Winter | Spring | Summer | Autumn |
|---|---|---|---|---|---|---|
| $M_2$ | Amplitude | 73.97 | 67.29 | 71.56 | 80.87 | 76.04 |
| | Phase lag | 210.03 | 209.11 | 209.42 | 212.04 | 209.52 |
| $S_2$ | Amplitude | 20.81 | 19.83 | 18.87 | 22.46 | 22.06 |
| | Phase lag | 273.74 | 276.69 | 272.81 | 279.67 | 265.80 |
| $K_1$ | Amplitude | 30.30 | 27.96 | 29.99 | 32.09 | 31.13 |
| | Phase lag | 30.57 | 31.31 | 30.39 | 33.55 | 27.01 |
| $O_1$ | Amplitude | 24.43 | 21.92 | 24.13 | 25.37 | 26.27 |
| | Phase lag | 345.91 | 351.35 | 340.62 | 348.00 | 343.79 |

5   **Table 2. Averaged amplitudes (cm) and phase lag (°) of the principal tidal constituents obtained using EHA at Dalian**

| Constituents | Parameter | Annual | Winter | Spring | Summer | Autumn |
|---|---|---|---|---|---|---|
| $M_2$ | Amplitude | 97.32 | 97.53 | 96.79 | 97.96 | 97.00 |
| | Phase lag | 54.25 | 52.93 | 53.43 | 55.34 | 55.28 |
| $S_2$ | Amplitude | 30.89 | 32.54 | 30.91 | 29.1 | 31.02 |
| | Phase lag | 100.12 | 99.45 | 102.47 | 101.08 | 97.44 |
| $K_1$ | Amplitude | 25.32 | 25.02 | 24.78 | 26.18 | 25.3 |
| | Phase lag | 240.94 | 242.44 | 239.91 | 239.31 | 242.15 |
| $O_1$ | Amplitude | 17.91 | 17.41 | 18.05 | 17.88 | 18.29 |
| | Phase lag | 210.29 | 212.4 | 208.54 | 209.62 | 210.65 |

**Table 3. Model settings for the numerical experiments**

| No. | Season | $A_z$ [a] (m²/s) | Depth (m) |
|---|---|---|---|
| Exp1 | Winter | $2.0 \times 10^{-3}$ | Original |
| Exp2 | Summer | $2.0 \times 10^{-3}$ | Original |
| Exp3 | Summer | $1.0 \times 10^{-3}$ | Original |
| Exp4 | Summer | $2.0 \times 10^{-3}$ | Original+0.2 |

[a] Vertical eddy viscosity coefficient.

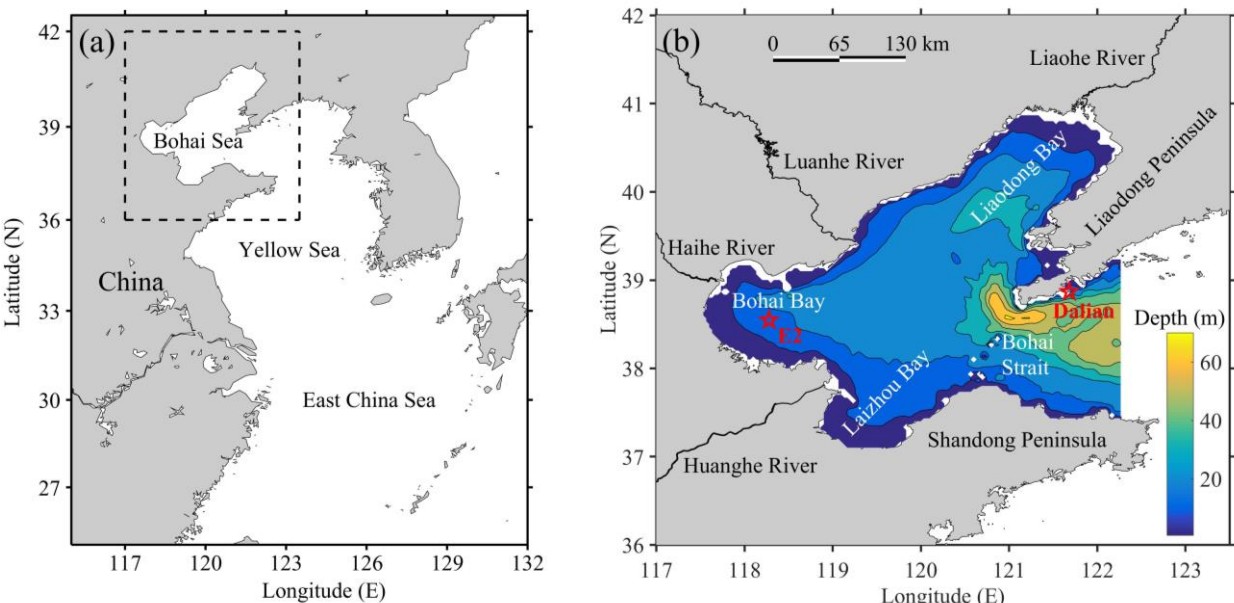

**Figure 1. (a) General location of the Bohai Sea (rectangle with dashed lines); and (b) locations of the observation stations (red stars), E2 and Dalian, in the Bohai Sea, and bathymetry of the Bohai Sea (colours).**

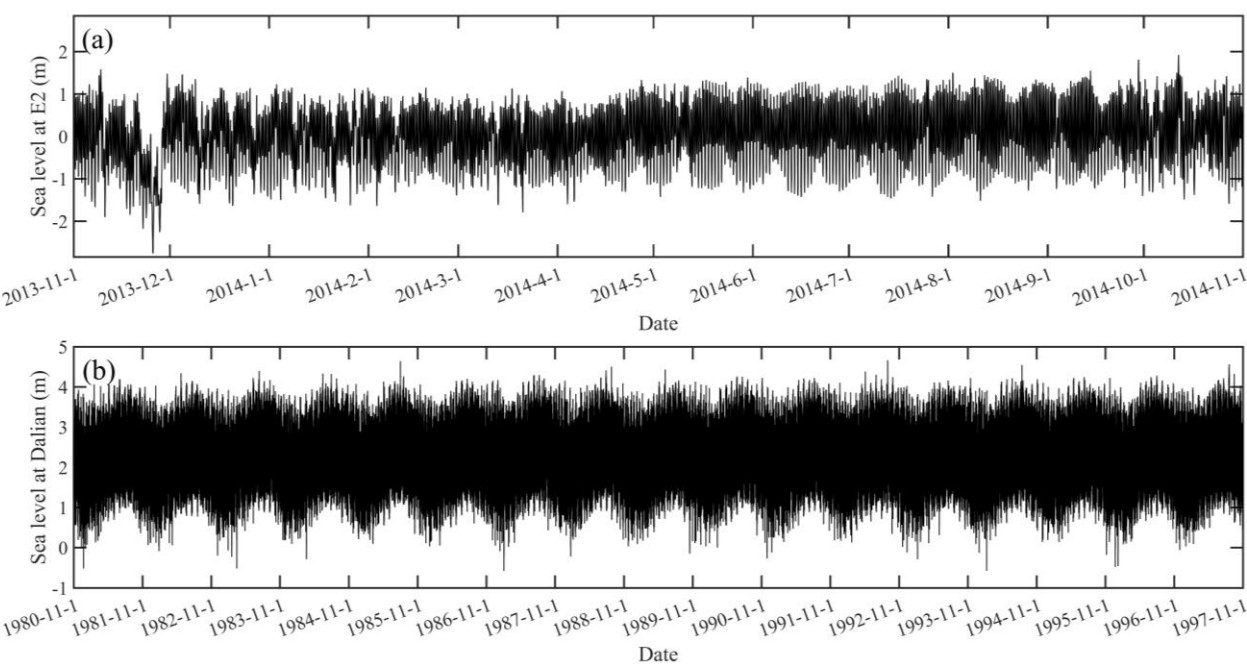

**Figure 2. Time series of the observed sea level at (a) E2 and (b) Dalian.**

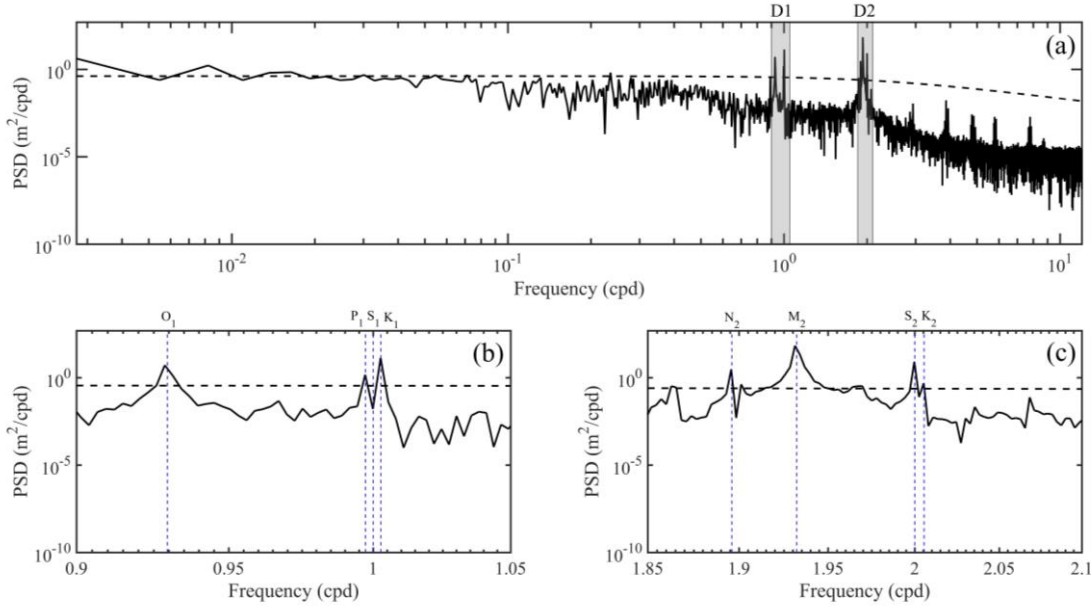

**Figure 3. Power spectral densities of the observed sea level at E2 (black line) in (a) all frequency bands, (b) the diurnal frequency band, and (c) the semidiurnal frequency band. In all panels, black dashed lines denote the corresponding 5% significance level against red noise.**

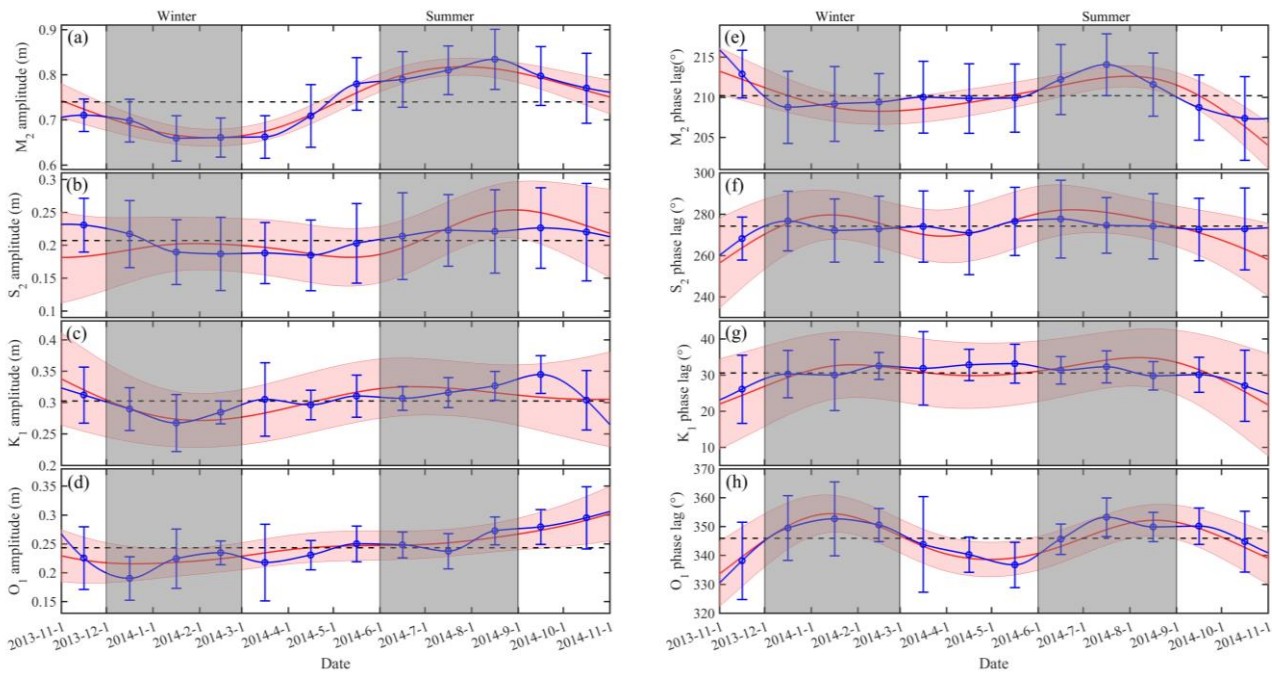

**Figure 4. Time series of the temporally varying tidal amplitudes of principal tidal constituents (a) $M_2$, (b) $S_2$, (c) $K_1$ and (d) $O_1$ at E2 estimated with CHA (black dashed lines), SHA (blue lines) and EHA (red lines). (e-h) Similar to (a-d), but**

**for the estimated temporally varying tidal phase lags. Blue vertical bars and pink shading indicate the corresponding 95% confidence intervals.**

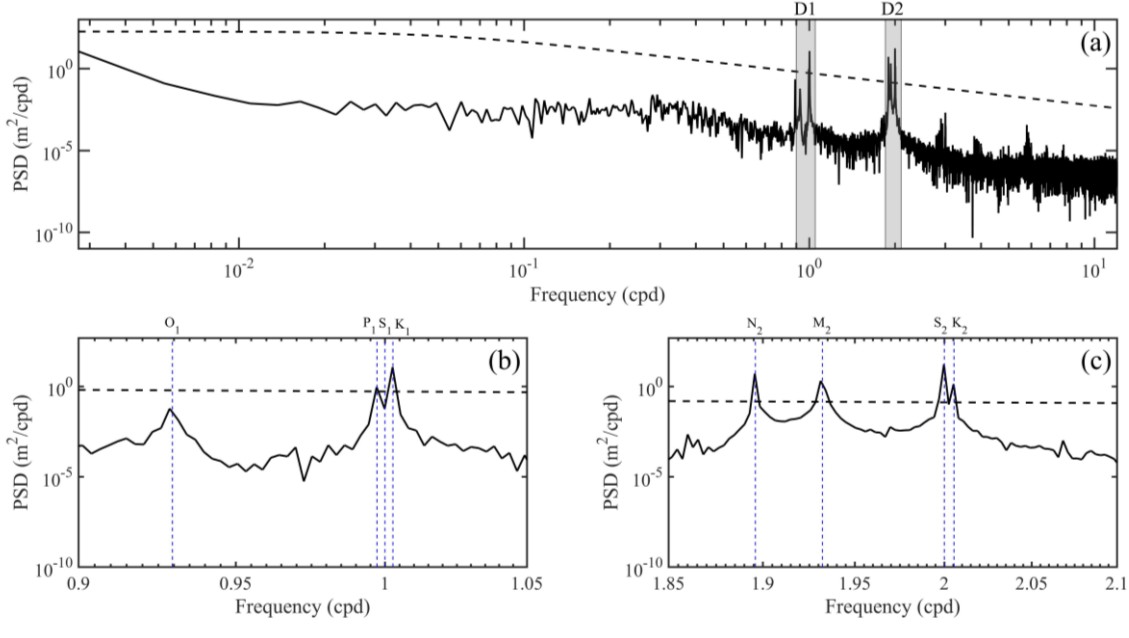

**Figure 5. Similar to Figure 3, but for those at Dalian.**

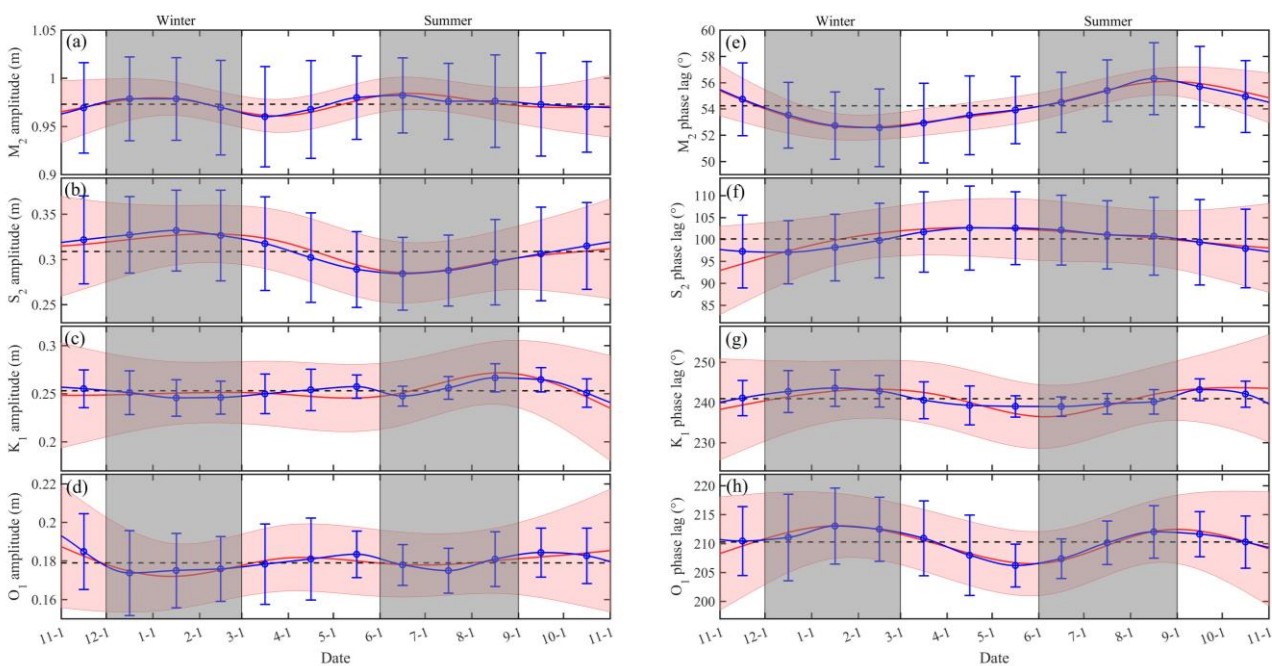

**Figure 6. Similar to Figure 4, but for multi-yearly averaged values at Dalian.**

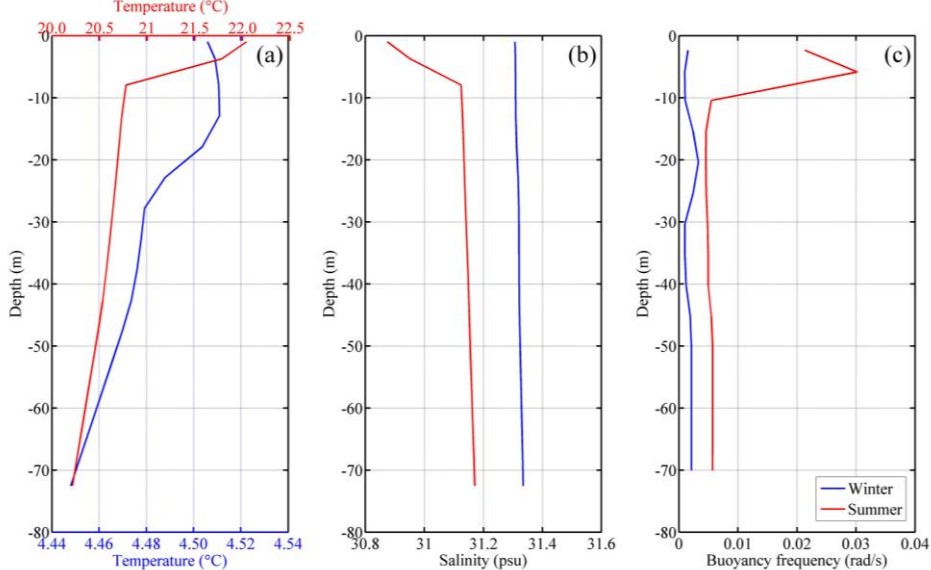

**Figure 7. Horizontally homogeneous profiles of the initial (a) temperature, (b) salinity and (c) buoyancy frequency used in the numerical experiments, in winter (blue solid lines) and in summer (red solid lines).**

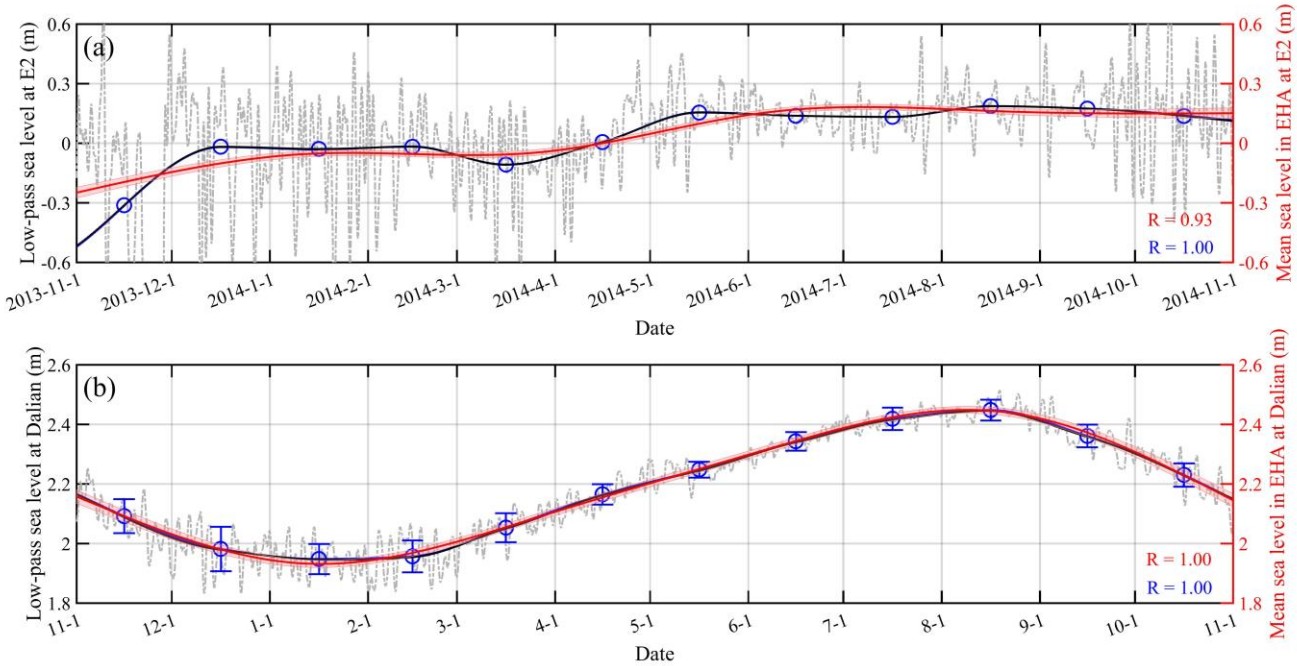

**Figure 8. Time series of the original low-pass sea level (grey line), their monthly averages (blue circles), the interpolated values of the monthly averages using the cubic spline interpolation (blue line), and the estimated mean sea level using**

SHA (black line) and EHA (red line), at (a) E2 and (b) Dalian. Only the original low-pass sea levels with absolute values less than 0.6 m are shown in panel (a). Pink shading indicates the corresponding 95% confidence intervals, while blue vertical bars designate the standard deviation in multi-yearly averaging.

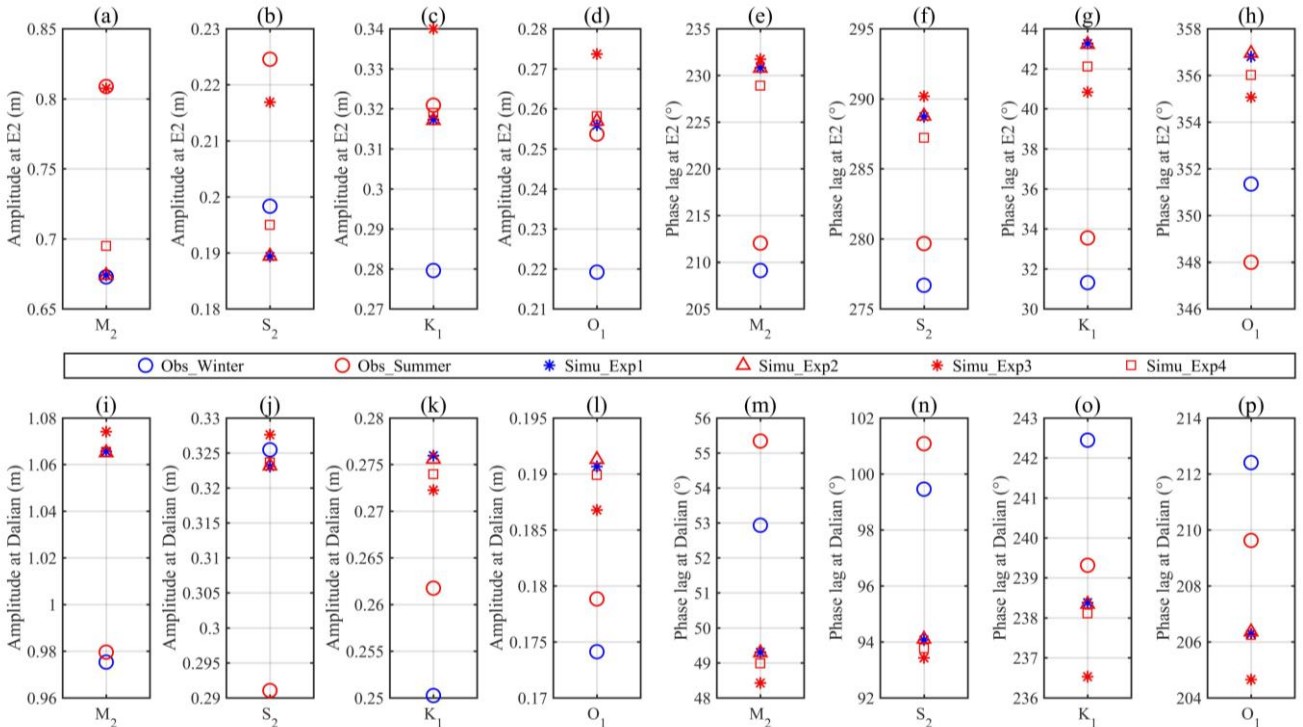

Figure 9. (a) Averaged $M_2$ amplitude in winter (blue circle) and summer (red circle) by analysing observations at E2 using EHA, and those obtained by analysing the simulated results in Exp1 (blue asterisk), Exp2 (red triangle), Exp3 (red asterisk) and Exp4 (red square). (b-d) Similar to (a), but for $S_2$, $K_1$ and $O_1$ at E2, respectively. (e-h) Similar to (a-d), but for the phase lags at E2. (i-p) Similar to (a-h), but for those at Dalian.

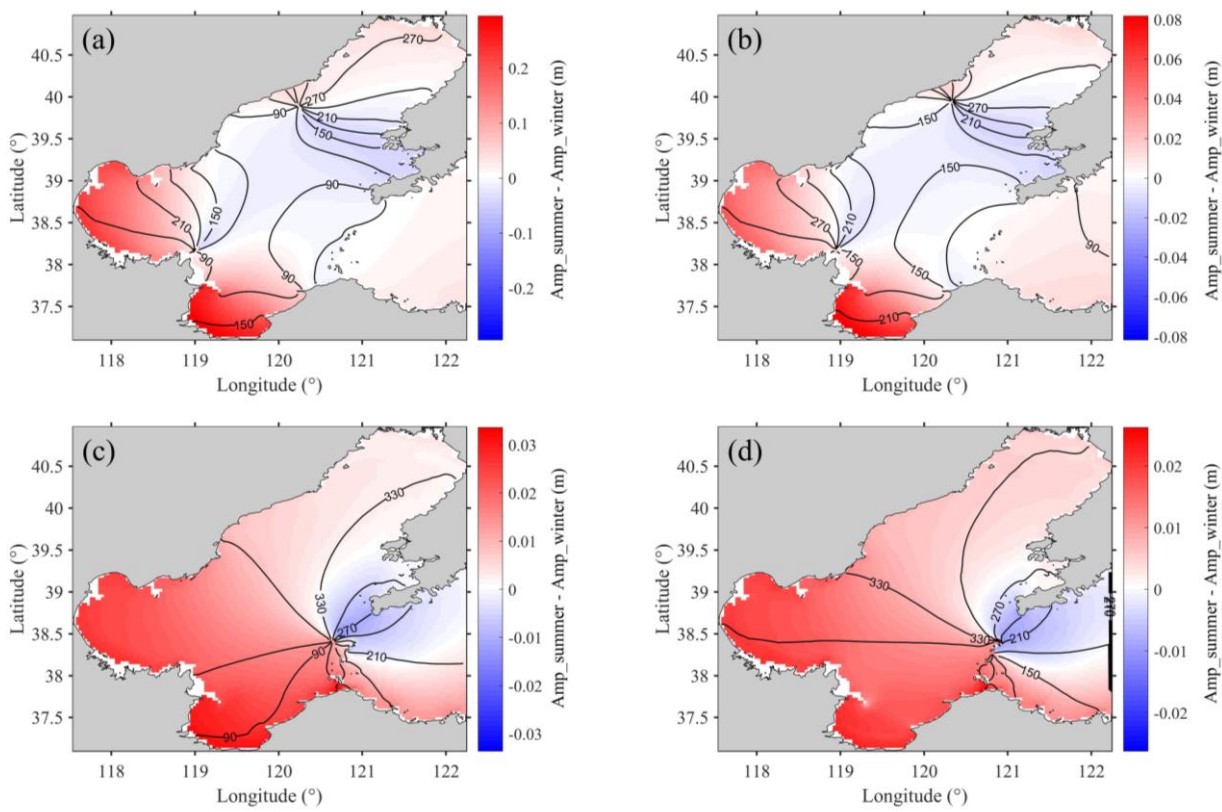

**Figure 10. (a) Difference between the simulated M$_2$ amplitudes in summer (Exp3) and those in winter (Exp1) (colours), and the co-phase lines of the M$_2$ tide in winter (Exp1) (black lines). (b-d) similar to (a), but for S$_2$, K$_1$ and O$_1$, respectively.**