# Peer review of "Seasonal variation of the principal tidal constituents in the Bohai Sea"

_Ocean Science, 2019_

## Referee Comment (RC1) · Anonymous Referee #1 · 12 Jun 2019

OVERVIEW -This study provides an analysis of the seasonal variation (or as I have called it in the past, "seasonality") of the M2, S2, K1, and O1 tides in Bohai Bay. -This is a good an important subject to study, as the seasonal variability of tides (that is not astronomical) may be a significant part of water level variability in certain regions, particularly those subject to strongly seasonal weather patterns (like the monsoons in Asia), or especially shallow regions (like the Gulf of Thailand). -Overall, I think the methods and approach is sound, and the numerical models are sound. I would really like to accept this paper, but I am concerned that the missing data is a big limitation to the trust ability of the results, and this should at least be discussed more. -I do have some other major concerns, listed below, and some other minor comments. -One factor that will have to be addressed in this manuscript is the English writing quality. It is

not so bad, but it also not so good, and I can notice a number of small errors and style points (such as too many dependent clauses beginning sentences or paragraphs) that should be addressed to make this a better paper. So I would highly recommend having a native English speaker give this paper a very close read before acceptance. -As far as the "Enhanced harmonic analysis" methods and claims of novelty... There have been multiple improvements to T\_TIDE in the past decades, such a R\_T\_TIDE (Leffler and Jay, 2009), "versatile" tidal analysis (Foreman, 2009) and U\_TIDE (Codiga, 2011). Were any of these methods tried in addition to T TIDE? If you haven't tried these, then it's harder to claim that your method is "enhanced" more than T\_TIDE when others have already produced "enhanced" methods. How does your method improve on all these past approaches? -While one year of hourly water level data is indeed enough to resolve most tides and reveal a seasonal cycle, I have some reservations about how much you can conclude about the seasonality cycle based on one year of observations. It is likely somewhat constant year-by-year, but it is hard to be sure. If you look at Devlin et al, 2018, for example, they looked at 30 years of data to show the seasonal cycle was basically constant, but not identical. If a single year had some sort of rare event (like a particularly strong storm season), this might skew the seasonal pattern a bit. I understand that one year is guite a bit of data to get from mooring, and more data is likely not available, but this is still a limitation that should be discussed somewhere in the paper. -Also, the existence of gaps in the single year dataset is also unfortunate, because it is hard to know the effect of the gaps without comparing them to a full year's data. Are there any ground-based tide gauges nearby to compare to? -I think that the data gaps at E1 are too extensive ti trust this location. Unless you have another nearby location (like a ground-based tide gauge) to compare to, I don't think that these results can be trusted. So maybe this location should be omitted. -My recommendation is for major revision, with more discussions about the effects of missing data, and the comparisons to other methods and the results of other who have analysed seasonality.

COMMENTS: (I really wish there were continuous line numbers, but since there are not, I will index comments by section and/or page number and line number within the

page) -ABSTRACT: -I understand what you are saying in your opening, and I do agree with everything you say, but the English is somewhat awkward right out of the gate, and this will confuse other readers who are not as familiar with tides. The text here just needs some minor refinement -INTRODUCTION -Page 2, line 3-4: Devlin et al, 2018 also looked at K1, O1, and S2 seasonality found some interesting patterns of seasonality in K1 and O1 at some locations, though M2 was the primary seasonal variation observed. -Line 5: "Several other studies have analysed the seasonal variability..." -Line 7: "tidal constants"  $\rightarrow$  "tidal constituents" -Line 14: "major tidal harmonic parameters"  $\rightarrow$  "largest tidal constituents"

-OBSERVATIONS AND METHODS -Page 2, line 26: How about eustatic sea level rise? -Page 3, line 4: See comment above about improvements to T TIDE

Section 2.3 -I am bit unclear about what you mean by SHA here. Do you mean that you are harmonically analysing datasets that are monthly or shorter? If so, then you will resolve the four major tides, but you will likely not constrain the natural yearly variability that one-year analyses contain (such as SA and SSA). Therefore, any seasonality you observed might actually be just an artefact of the mathematics. If you had more than one year of data, you could perform overlapping one-year HA at one-month steps, then any seasonality revealed would be more "real" Maybe I am missing something, but at least you could explain it better. In any case, this is the obvious issue with only using one single year of data. -Also, 15 days of data within a month as a criteria makes the results much worse, and could be highly spurious. This is less than 50% completeness, while I believe a criteria of 75% to 80% is needed

-RESULTS -Page 5, Line 4: Of course gaps will influence results, especially since you have such a relaxed criterion of completeness (50%) and such a short time-series ( $\sim$  1year). This has to be discussed more, and perhaps there is just not enough data to perform this study adequately.

-MECHANISMS -Seasonal variations of sea level are indeed important and there can

СЗ

be a lot of reasons (monsoons, etc.), but can you discuss more what causes these variations, and how these might influence tides physically? Line 14-15: I think that the importance of sea ice in the reason could still be important to seasonality, if you consider the "back-effect" connection of coastal embayments and open water as discussed by Arbic and Garret, 2010; Arbic, 2009 via resonance mechanisms. So, even if the ice cover is far away from your observations site, it could still be important. These studies should at least be mentioned and discussed here. -Page 8, line 14: "... as the differences... larger than 0." Is kind of an elementary statement, you can omit this

-Page 6, Line 16-19: Can you restate this as a statement instead of a question? It doesn't read well in the middle of the paragraph as written.

---

## Referee Comment (RC2) · Anonymous Referee #2 · 9 Jul 2019

please see supplement.

Please also note the supplement to this comment:
https://www.ocean-sci-discuss.net/os-2019-43/os-2019-43-RC2-supplement.pdf

---

## Author Comment (AC2) · 17 Aug 2019

Please find our responses to comments by Reviewer#2 in the attached file.

Please also note the supplement to this comment:
https://www.ocean-sci-discuss.net/os-2019-43/os-2019-43-AC2-supplement.pdf

––––––––––––––––––––––––––––

---

## Author Response (AR1)

Dear Editor,

We'd like to thank both reviewers for their comments and positive appraisal. Based on their comments, we have changed the manuscript. Please find below a point-to-point response marked in blue, with changes in the text cited in green. The page numbers and line numbers in this document refer to the revised version of the manuscript without using the revision mode in the Microsoft Word.

**Response to comments by Reviewers #1**

(1) OVERVIEW -This study provides an analysis of the seasonal variation (or as I have called it in the past, "seasonality") of the M2, S2, K1, and O1 tides in Bohai Bay. -This is a good an important subject to study, as the seasonal variability of tides (that is not astronomical) may be a significant part of water level variability in certain regions, particularly those subject to strongly seasonal weather patterns (like the monsoons in Asia), or especially shallow regions (like the Gulf of Thailand). -Overall, I think the methods and approach is sound, and the numerical models are sound. I would really like to accept this paper, but I am concerned that the missing data is a big limitation to the trust ability of the results, and this should at least be discussed more. I do have some other major concerns, listed below, and some other minor comments.

**Reply:**

Thanks a lot for the positive assessment and constructive comments of our paper.

According to your fifth comment and the third comments from Reviewer #2, we have omitted the E1 station, at which there are some gaps in the sea level observations; in addition, the tidal gauge station Dalian, at which there are 17-year sea-level observations, is added to investigate the seasonal variation of principal tidal constituents in the Bohai Sea. Therefore, we have modified the title of the manuscript to be 'Seasonal variation of the principal tidal constituents in the Bohai Sea'.

**Changes:**

P1L1: Seasonal variation of the principal tidal constituents in the Bohai Sea

P2L24: Hourly sea-level data at the Dalian tidal gauge station were obtained from the

University of Hawaii Sea Level Center and used. After 1979, Dalian shared position with Laohutan (Feng et al., 2015), so the sea-level data at Dalian were comprised of data from Dalian from 1980-1990 and from Laohutan from 1991-1997, as shown in Figure 2b.

(2) -One factor that will have to be addressed in this manuscript is the English writing quality. It is not so bad, but it also not so good, and I can notice a number of small errors and style points (such as too many dependent clauses beginning sentences or paragraphs) that should be addressed to make this a better paper. So I would highly recommend having a native English speaker give this paper a very close read before acceptance.

**Reply:**

Thank you for your constructive suggestion. We are sorry that the old manuscript was hurriedly written and the English writing was not good. We have carefully modified and employed the professional language editors of Elsevier Language Editing Services to edit the revised manuscript.

The language editing certification is attached in the end of this document.

(3) As far as the "Enhanced harmonic analysis" methods and claims of novelty... There have been multiple improvements to T\_TIDE in the past decades, such a R\_T\_TIDE (Leffler and Jay, 2009), "versatile" tidal analysis (Foreman, 2009) and U\_TIDE (Codiga, 2011). Were any of these methods tried in addition to T\_TIDE? If you haven't tried these, then it's harder to claim that your method is "enhanced" more than T\_TIDE when others have already produced "enhanced" methods. How does your method improve on all these past approaches?

**Reply:**

Thank you for your comment.

The R\_T\_TIDE (Leffler and Jay, 2009), versatile tidal analysis (Foreman et al., 2009) and U\_TIDE (Codiga, 2011), are indeed improved to T\_TIDE in different perspectives. According to Equation (8) in Leffler and Jay (2009), Equation (4) in Forman et al. (2009) and Equation (2) in Codiga (2011), it can be found that the harmonic parameters (i.e., amplitude and phase lag)

are assumed to be constant. In fact, the harmonic parameters are not constant and have multiscale temporal variations, as shown in Corkan (1934), Kang et al. (1995), Müller et al. (2014), Devlin et al. (2018), and so on.

Jin et al. (2018) and Pan et al. (2018) developed the enhanced harmonic analysis, in which the harmonic parameters (i.e., amplitude and phase lag) are assumed to be temporally varying and computed directly within the least squares fit. In this study, the nodal and astronomical argument corrections are embedded in the least squares fit; in addition, the harmonic parameters of minor constituents are assumed to be constant and computed together with the temporally varying harmonic parameters of principal tidal constituents to resolve more constituents and remain computational stability.

Although in the nonstationary tidal analysis tool NS\_TIDE (Matte et al., 2013) the harmonic parameters are also assumed to be temporally varying, the temporally varying harmonic parameters are assumed to be functions of river flow and greater diurnal tidal range at the reference station. Therefore, it just can be applied to the river tides. On the contrary, the enhanced harmonic analysis can be applied in analyzing any time series.

Therefore, the enhanced harmonic analysis used in this study improves on all the past approaches, which has been discussed in the revised manuscript.

Changes:

P9L16: In this study, the EHA developed in Jin et al. (2018) and Pan et al. (2018b) was further improved in order to resolve more tidal constituents by adding the minor constituents whose harmonic parameters were assumed to be constant and computed together with the temporally varying harmonic parameters of the principal tidal constituents. The nodal and astronomical argument corrections were embedded into the least square fit to eliminate the influences of nodal cycle and linearly varying astronomical argument. In fact, there have been multiple improvements to T\_TIDE in the past decades, such as R\_T\_TIDE (Leffler and Jay, 2009), versatile tidal analysis (Foreman et al., 2009), U\_TIDE (Codiga, 2011) and NS\_TIDE (Matte et al., 2013). In R\_T\_TIDE, versatile tidal analysis and U\_TIDE, the harmonic parameters (i.e., amplitude and phase lag) are assumed to be constant, although they have some

improvements to T\_TIDE. However, harmonic parameters are not constant and have multiscale temporal variations, as shown in Corkan (1934), Kang et al. (1995), Müller et al. (2014), Devlin et al. (2018), and so on. Neglecting seasonal variation of tides will introduce significant error in sea-level prediction (Fang and Wang, 1986). Therefore, EHA, in which the harmonic parameters of the principal tidal constituents are assumed to be temporally varying and computed directly within the least squares fit, improved T\_TIDE in other ways. In NS\_TIDE, the harmonic parameters are also assumed to be temporally varying. However, the temporally varying harmonic parameters are assumed to be functions of river flow and greater diurnal tidal range at the reference station, so it can be applied only to river tides, while EHA can be applied in analyzing any time series. On the whole, EHA used in this study is indeed enhanced than other methods.

(4) While one year of hourly water level data is indeed enough to resolve most tides and reveal a seasonal cycle, I have some reservations about how much you can conclude about the seasonality cycle based on one year of observations. It is likely somewhat constant year-by-year, but it is hard to be sure. If you look at Devlin et al, 2018, for example, they looked at 30 years of data to show the seasonal cycle was basically constant, but not identical. If a single year had some sort of rare event (like a particularly strong storm season), this might skew the seasonal pattern a bit. I understand that one year is quite a bit of data to get from mooring, and more data is likely not available, but this is still a limitation that should be discussed somewhere in the paper.

**Reply:**

Thank you for your constructive suggestion.

The duration of hourly sea-level observations at E2 used in this study was only one year, which is a limitation. Although the rare event might slightly skew seasonal pattern of tides, the seasonal variations of the principal tidal constituents  $M_2$ ,  $S_2$ ,  $K_1$  and  $O_1$  obtained using EHA were the same as those using traditional EHA, indicating that the results are not unreasonable and can reflect the seasonal variations of tides in the analysis period. The strong seasonal variations of the principal tidal constituents can be captured by the results of numerical experiments.

Besides, according to your fifth comment and the third comments from Reviewer #2, we have omitted the E1 station, at which there are some gaps in the sea level observations; in addition, the tidal gauge station Dalian, at which there are 17-year sea-level observations, is added to investigate the seasonal variation of principal tidal constituents in the Bohai Sea. Changes:

P10L1: The duration of hourly sea-level observations at E2 used in this study was only one year, which is a limitation. Although the rare event might slightly skew seasonal pattern of tides, the seasonal variations of the principal tidal constituents  $M_2$ ,  $S_2$ ,  $K_1$  and  $O_1$  obtained using EHA were the same as those using traditional EHA, indicating that the results are not unreasonable and can reflect the seasonal variations of tides in the analysis period. The strong seasonal variations of the principal tidal constituents can be captured by the results of numerical experiments. The multi-annually averaged results at Dalian also showed the seasonal variations of the principal tidal constituents, but the results of numerical experiments were not in accordance with the observed results, which may be because only the horizontally homogeneous profiles of the initial temperature and salinity were used and the temporally varying ocean circulation were not considered.

(5) Also, the existence of gaps in the single year dataset is also unfortunate, because it is hard to know the effect of the gaps without comparing them to a full year's data. Are there any ground-based tide gauges nearby to compare to? -I think that the data gaps at E1 are too extensive ti trust this location. Unless you have another nearby location (like a ground-based tide gauge) to compare to, I don't think that these results can be trusted. So maybe this location should be omitted.

**Reply:**

Thank you for your constructive suggestion. We are sorry that there are no ground-based tide gauge stations nearby to compare to. The gaps at E1 are indeed difficultly evaluated, as you

said. According to your suggestion, we have omitted E1 station.

Besides, the tidal gauge station Dalian, at which there are 17-year sea-level observations, is added to investigate the seasonal variation of the principal tidal constituents in the Bohai Sea. Therefore, both the mooring data and the tidal gauge station data are used in the revised manuscript.

(6) My recommendation is for major revision, with more discussions about the effects of missing data, and the comparisons to other methods and the results of other who have analysed seasonality.

**Reply:**

Thank you for giving us the opportunity to further improve the old manuscript.

According to the suggestions from you and Reviewer #2, we omit the E1 station, at which there are some gaps in the sea level observations; in addition, the tidal gauge station Dalian, at which there are 17-year sea-level observations, is added to investigate the seasonal variation of the principal tidal constituents in the Bohai Sea. Because there are no large gaps in the observations at both E2 and Dalian, the effect of missing data is not discussed in the revised manuscript.

The comparisons to other methods, including R\_T\_TIDE (Leffler and Jay, 2009), versatile tidal analysis (Foreman et al., 2009), U\_TIDE (Codiga, 2011) and NS\_TIDE (Matte et al., 2013), have been added in section 5 (discussions) of the revised manuscript, which have been shown in the 'Changes' of the third comment.

The comparisons to other studies have been added in the revised manuscript, as shown in the following 'Changes'.

**Changes:**

P2L8: Previous studies have primarily focused on the seasonal variation in the  $M_2$  amplitude without considering the seasonality of other tidal constituents and their phase lags (Gräwe et al., 2014). Indeed, several studies have investigated the seasonality of several constituents. For example, Fang and Wang (1986) studied the seasonal variations of  $M_2$ ,  $N_2$ ,  $O_1$

and  $M_4$  in the Bohai Sea by introducing astro-meteorological constituents; Devlin et al. (2018) found that the diurnal ( $K_1$  and  $O_1$ ) and semidiurnal ( $M_2$  and  $S_2$ ) tidal amplitudes and phase lags exhibited strong seasonal variability in the seas of Southeast Asia. In this study, sea-level observations at one mooring station (E2) and one tidal gauge station (Dalian) in the Bohai Sea were used to investigate the seasonal variability of the principal tidal constituents  $M_2$ ,  $S_2$ ,  $K_1$ and  $O_1$  using an enhanced harmonic analysis (EHA)..

P5L10: Compared to the annual averages, the mean  $M_2$  amplitude increased by 6.90 cm (approximately 9.33%) in the summer and decreased by 6.68 cm (approximately 9.03%) in the winter, close to the estimated values in Foreman et al. (1995) (6%), Huess and Andersen (2001) (6%) and Müller et al. (2014) (5%–10%).

P9L11: The spatial distribution of the absolute differences between the  $M_2$  tidal amplitude in summer and that in winter was similar to that in Müller et al. (2014).

(7) -ABSTRACT: -I understand what you are saying in your opening, and I do agree with everything you say, but the English is somewhat awkward right out of the gate, and this will confuse other readers who are not as familiar with tides. The text here just needs some minor refinement.

**Reply:**

Thank you for pointing this out. According to this suggestion, we have modified the first sentence in the revised manuscript, as shown in 'Changes'.

**Changes:**

P1L10: The seasonal variation of tides plays a significant role in water level changes in coastal regions.

(8) -INTRODUCTION -Page 2, line 3-4: Devlin et al, 2018 also looked at K1, O1, and S2 seasonality found some interesting patterns of seasonality in K1 and O1 at some locations, though M2 was the primary seasonal variation observed.

**Reply:**

Thank you for pointing this out. According to this comment, we have rewritten it as shown in following 'Changes'.

**Changes:**

P2L11: Devlin et al. (2018) found that the diurnal ( $K_1$  and  $O_1$ ) and semidiurnal ( $M_2$  and  $S_2$ ) tidal amplitudes and phase lags exhibited strong seasonal variability in the seas of Southeast Asia.

(9) -Line 5: "Several other studies have analysed the seasonal variability...".

**Reply:**

Thank you for pointing this out. We agree with your comment and have modified it in the revised manuscript.

**Changes:**

P2L4: Several other studies have analysed the seasonal variability of the  $M_2$  tide in polar regions.

(10) -Line 7: "tidal constants" ---> "tidal constituents".

**Reply:**

Thank you for pointing this out. We agree with your comment and have modified it in the revised manuscript.

**Changes:**

P2L5: Kagan and Sofina (2010) showed that the seasonal variability of tidal constituents was widespread in the Arctic Ocean.

(11) -Line 14: "major tidal harmonic parameters" ---> "largest tidal constituents".

**Reply:**

Thank you for pointing this out. We agree with your comment and have modified it in the revised manuscript.

Changes:

P2L13: In this study, sea-level observations at one mooring station (E2) and one tidal gauge station (Dalian) in the Bohai Sea were used to investigate the seasonal variability of the principal tidal constituents  $M_2$ ,  $S_2$ ,  $K_1$  and  $O_1$  using an enhanced harmonic analysis (EHA).

**(12) -OBSERVATIONS AND METHODS -Page 2, line 26: How about eustatic sea level rise?Reply:**

Thank you for pointing this out. We are sorry that the eustatic sea level rise was not explicitly shown in Equation (1) in the old manuscript. In the revised manuscript, we have modified the Equation (1) as follows:

$$\begin{aligned} \zeta(t) &= \zeta_0 + \sum_{k=1}^K \left\{ f_k(t) A_k \cos \left[ V_k(t) + u_k(t) - g_k \right] \right\} + R(t) \\ &= \zeta_0 + \sum_{i=1}^{N_{\text{NR}}} \left\{ f_i(t) A_i \cos \left[ V_i(t) + u_i(t) - g_i \right] \right\} + R(t) + \\ &\sum_{j=1}^N \left\{ f_j(t) A_j \cos \left[ V_j(t) + u_j(t) - g_j \right] + \sum_{n=1}^{N_1} f_n(t) A_n \cos \left[ V_n(t) + u_n(t) - g_n \right] \right\} \end{aligned}$$

where  $\zeta(t)$  is the total sea level;  $\zeta_0$  is the mean sea level; A and g are the amplitude and phase lag (UTC time, the same below), respectively; f and u are the nodal corrections to the amplitude and phase, respectively; V is the astronomical argument; R is the nontidal component; K is the number of tidal constituents;  $N_{\text{NR}}$  is the number of non-reference constituents;  $N_{\text{R}}$  is the number of reference constituents; and  $N_{\text{I}}$  is the number of constituents to be inferred from the  $j^{\text{th}}$  reference constituent.

Therefore, the eustatic sea level rise is included in R(t).

Changes:

P2L28:

A sea level is composed of components from different sources (Godin, 1972; Foreman, 1977; Fang, 1986; Pawlowicz et al., 2002; Foreman et al., 2009):

$$\begin{aligned} \zeta(t) &= \zeta_0 + \sum_{k=1}^{K} \left\{ f_k(t) A_k \cos \left[ V_k(t) + u_k(t) - g_k \right] \right\} + R(t) \\ &= \zeta_0 + \sum_{i=1}^{N_{\text{NR}}} \left\{ f_i(t) A_i \cos \left[ V_i(t) + u_i(t) - g_i \right] \right\} + R(t) + \\ &\sum_{j=1}^{N_{\text{R}}} \left\{ f_j(t) A_j \cos \left[ V_j(t) + u_j(t) - g_j \right] + \sum_{n=1}^{N_{\text{I}}} f_n(t) A_n \cos \left[ V_n(t) + u_n(t) - g_n \right] \right\} \end{aligned}$$
(1)

where  $\zeta(t)$  is the total sea level;  $\zeta_0$  is the mean sea level; A and g are the amplitude and phase lag (UTC time, the same below), respectively; f and u are the nodal corrections to amplitude and phase lag, respectively; V is the astronomical argument; R is the nontidal component; K is the number of tidal constituents;  $N_{\rm NR}$  is the number of non-reference constituents;  $N_{\rm R}$  is the number of reference constituents; and  $N_{\rm I}$  is the number of constituents to be inferred from the *j*th reference constituent.

(13) Page 3, line 4: See comment above about improvements to T\_TIDE.

**Reply:**

Thank you for your constructive suggestion.

As shown in the reply of the third comment, the R\_T\_TIDE (Leffler and Jay, 2009), versatile tidal analysis (Foreman et al., 2009) and U\_TIDE (Codiga, 2011), are indeed improved to T\_TIDE in different perspectives. According to Equation (8) in Leffler and Jay (2009), Equation (4) in Forman et al. (2009) and Equation (2) in Codiga (2011), it can be found that the harmonic parameters (i.e., amplitude and phase lag) are assumed to be constant. In fact, the harmonic parameters are not constant and have multiscale temporal variations, as shown in Corkan (1934), Kang et al. (1995), Müller et al. (2014), Devlin et al. (2018), and so on.

Jin et al. (2018) and Pan et al. (2018) developed the enhanced harmonic analysis, in which the harmonic parameters (i.e., amplitude and phase lag) are assumed to be temporally varying and computed directly within the least squares fit. In this study, the nodal and astronomical argument corrections are embedded in the least squares fit; in addition, the harmonic parameters of minor constituents are assumed to be constant and computed together with the temporally varying harmonic parameters of principal tidal constituents to resolve more constituents and

**remain computational stability.**

(14) Section 2.3 -I am bit unclear about what you mean by SHA here. Do you mean that you are harmonically analysing datasets that are monthly or shorter? If so, then you will resolve the four major tides, but you will likely not constrain the natural yearly variability that one-year analyses contain (such as SA and SSA). Therefore, any seasonality you observed might actually be just an artefact of the mathematics. If you had more than one year of data, you could perform overlapping one-year HA at one-month steps, then any seasonality revealed would be more "real" Maybe I am missing something, but at least you could explain it better. In any case, this is the obvious issue with only using one single year of data.

**Reply:**

Thank you for your comments.

In SHA, the sea level observations are divided into monthly segments according to calendar month and CHA with nodal and inference corrections is applied to each monthly segment to obtain the discrete tidal harmonic parameters (i.e., amplitude and phase lag), which has been used in many previous studies (e.g., Foreman et al., 1995; Kang et al., 1995; Müller et al., 2014; Devlin et al., 2018). Then the discrete amplitude and phase lag at each monthly segment are interpolated using cubic spline interpolation to obtain the temporally varying amplitude and phase lag.

It is reasonable to use the monthly analysis to obtain the seasonal variability of principal tide constituents, as long as a valid inference procedure is used. The seasonality in principal tide constituents have been investigated using monthly analysis in many previous studies (e.g., Foreman et al., 1995; Kang et al., 1995; Müller et al., 2014; Devlin et al., 2018).

As pointed out by Reviewer #2, the unresolved constituents in the monthly analysis will cause the spurious seasonality in  $K_1$  and  $S_2$  in the old manuscript. Following the suggestions from you and Reviewer #2, we have redone the data analysis in the revised manuscript. (1) Consider the nodal correction and infer unresolved constituents in SHA using the function in T\_TIDE; (2) Embed the nodal correction into EHA to eliminate the influence of 18.6-year nodal

modulation and add the minor constituents, whose harmonic parameters are assumed to be constant, to resolve more constituents. The seasonal variations of the principal tidal constituents at E2 and Dalian are shown in Figures 4 and 6, respectively, in the revised manuscript. The unreasonably large semi-annual modulations in  $S_2$  and  $K_1$  have disappeared and the results are reasonable on the whole.

**Changes:**

P3L13: Following Foreman et al. (1995), Kang et al. (1995), Müller et al. (2014) and Devlin et al. (2018), sea-level observations are divided into monthly segments by calendar month and CHA with nodal and inference corrections is applied to each monthly segment to obtain the discrete tidal harmonic parameters (i.e., amplitude and phase lag). Then the discrete amplitude and phase lag at each monthly segment are interpolated using cubic spline interpolation to obtain the temporally varying amplitudes and phase lags. This methodology is termed segmented harmonic analysis (SHA). Following Kang et al. (1995), the sea-level observations in every monthly segment are analyzed only when the duration of the observations is greater than 26 days.

(15) -Also, 15 days of data within a month as a criteria makes the results much worse, and could be highly spurious. This is less than 50% completeness, while I believe a criteria of 75% to 80% is needed.

**Reply:**

Thank you for your constructive suggestions. Following this comment and Kang et al. (1995), we have modified the criteria from 15 days to 26 days in the revised manuscript. Changes:

P3L17: Following Kang et al. (1995), the sea-level observations in every monthly segment are analyzed only when the duration of the observations is greater than 26 days.

(16) RESULTS -Page 5, Line 4: Of course gaps will influence results, especially since you have such a relaxed criterion of completeness (50%) and such a short time-series (~ 1year). This has to be discussed more, and perhaps there is just not enough data to perform this study adequately. **Reply:**

Thank you for your constructive suggestion. According to your suggestion and those from Reviewer #2, we have omitted the sea-level observations at E1 and added the multiyear sealevel observations at Dalian tidal gauge station. At both E2 and Dalian, there were no large gaps in the data, as shown in Figure 2 in the revised manuscript.

(17) MECHANISMS -Seasonal variations of sea level are indeed important and there can be a lot of reasons (monsoons, etc.), but can you discuss more what causes these variations, and how these might influence tides physically?

**Reply:**

Thank you for your constructive suggestions.

According to this suggestion, we have added the discussions of season variations of stratification and tides through physical processes. In winter, the strong northwest Asia monsoon develops a vertically will-mixed condition (Yanagi et al., 2001; Jeon et al., 2014), which will not stabilize the tidal currents and will lose more energy, leading to smaller tidal amplitudes. As the surface heating rate and freshwater discharge increase, the mixing is insufficient to homogenize the input potential energy and cause stratified conditions in summer (Huang et al., 1999; van Haren, 2000), during which the reduced vertical eddy viscosity will increase the tidal amplitudes. However, the  $S_2$  amplitude at Dalian was larger in winter and smaller in summer, which is inconsistent with the other principal tidal constituents and should be further investigated in future studies.

**Changes:**

P10L9: The seasonal variations of stratification and vertical eddy viscosity and their influences on the tidal amplitudes may be as follows. In winter, the strong northwest Asia

monsoon develops a vertically will-mixed condition (Yanagi et al., 2001; Jeon et al., 2014), which will not stabilize the tidal currents and will lose more energy, leading to smaller tidal amplitudes. As the surface heating rate and freshwater discharge increase, the mixing is insufficient to homogenize the input potential energy and cause stratified conditions in summer (Huang et al., 1999; van Haren, 2000), during which the reduced vertical eddy viscosity will increase the tidal amplitudes. However, the  $S_2$  amplitude at Dalian was larger in winter and smaller in summer, which is inconsistent with the other principal tidal constituents and should be further investigated in future studies.

(18) Line 14-15: I think that the importance of sea ice in the reason could still be important to seasonality, if you consider the "back-effect" connection of coastal embayments and open water as discussed by Arbic and Garret, 2010; Arbic, 2009 via resonance mechanisms. So, even if the ice cover is far away from your observations site, it could still be important. These studies should at least be mentioned and discussed here.

**Reply:**

Thank you for your constructive suggestions. According to the back-effect connection of the coastal shelf and open ocean via resonance mechanisms (Arbic et al., 2009; Arbic and Garrett, 2010), sea ice may be important to the seasonality of principal tidal constituents. However, Zhang et al. (2019) developed a three-dimensional ice-ocean coupled model based on Finite Volume Community Ocean Model (FVCOM) and found that the damping effect of sea ice on the astronomical tides were almost negligible in the Bohai Sea using the numerical experiments.

According to your comment, we have discussed the influences of sea ice in the revised manuscript by citing Arbic et al. (2009), Arbic and Garret (2010) and Zhang et al. (2019). Changes:

P7L13: According to the back-effect connection of the coastal shelf and open ocean via resonance mechanisms (Arbic et al., 2009; Arbic and Garrett, 2010), sea ice may be important to the seasonality of principal tidal constituents. However, Zhang et al. (2019) found that the

damping effect of sea ice on the astronomical tides was almost negligible in the Bohai Sea employing numerical experiments with a three-dimensional ice-ocean coupled model. Therefore, ice coverage was not considered in this study.

(19) -Page 8, line 14: ".. as the differences ... larger than 0." is kind of an elementary statement, you can omit this.

**Reply:**

Thank you for pointing this out. According to your suggestions, we have deleted this sentence in the revised manuscript.

(20) -Page 6, Line 16-19: Can you restate this as a statement instead of a question? It doesn't read well in the middle of the paragraph as written.

**Reply:**

Thank you for pointing this out. According to your suggestions, we have deleted this sentence in the revised manuscript.

**References**

- Arbic, B. K. and Garrett, C.: A coupled oscillator model of shelf and ocean tides, Continental Shelf Research, 30, 564-574, 2010.
- Arbic, B. K., Karsten, R. H., and Garrett, C.: On tidal resonance in the global ocean and the back - effect of coastal tides upon open - ocean tides, Atmosphere-Ocean, 47, 239-266, 2009.
- Codiga, D. L.: Unified tidal analysis and prediction using the UTide Matlab functions, Graduate School of Oceanography, University of Rhode Island, Narragansett, RI, 2011.
- Corkan, R. H.: An Annual Perturbation in the Range of Tide, Proceedings of the Royal Society of London, 144, 537-559, 1934.

- Devlin, A. T., Zaron, E. D., Jay, D. A., Talke, S. A., and Pan, J.: Seasonality of Tides in Southeast Asian Waters, Journal of physical oceanography, 48, 1169-1190, 2018.
- Foreman, M. G., Cherniawsky, J., and Ballantyne, V.: Versatile harmonic tidal analysis: Improvements and applications, Journal of atmospheric and oceanic technology, 26, 806-817, 2009.
- Huang, D., Su, J., and Backhaus, J. O.: Modelling the seasonal thermal stratification and baroclinic circulation in the Bohai Sea, Continental Shelf Research, 19, 1485-1505, 1999.
- Jeon, C., Park, J. H., Varlamov, S. M., Yoon, J. H., Kim, Y. H., Seo, S., Park, Y. G., Min, H. S., Lee, J. H., and Kim, C. H.: Seasonal variation of semidiurnal internal tides in the East/Japan Sea, Journal of Geophysical Research: Oceans, 119, 2843-2859, 2014.
- Jin, G., Pan, H., Zhang, Q., Lv, X., Zhao, W., and Gao, Y.: Determination of Harmonic Parameters with Temporal Variations: An Enhanced Harmonic Analysis Algorithm and Application to Internal Tidal Currents in the South China Sea, Journal of atmospheric and oceanic technology, 35, 1375-1398, 2018.
- Kang, S. K., Chung, J.-y., Lee, S.-R., and Yum, K.-D.: Seasonal variability of the M 2 tide in the seas adjacent to Korea, Continental Shelf Research, 15, 1087-1113, 1995.
- Leffler, K. E. and Jay, D. A.: Enhancing tidal harmonic analysis: Robust (hybrid L1/L2) solutions, Continental Shelf Research, 29, 78-88, 2009.
- Müller, M., Cherniawsky, J. Y., Foreman, M. G., and von Storch, J.-S.: Seasonal variation of the M2 tide, Ocean Dynamics, 64, 159-177, 2014.
- Matte, P., Jay, D. A., and Zaron, E. D.: Adaptation of classical tidal harmonic analysis to nonstationary tides, with application to river tides, Journal of atmospheric and oceanic technology, 30, 569-589, 2013.
- Pan, H., Lv, X., Wang, Y., Matte, P., Chen, H., and Jin, G.: Exploration of tidal-fluvial interaction in the Columbia River estuary using S\_TIDE, Journal of Geophysical Research: Oceans, 123, 6598-6619, 2018.

- van Haren, H.: Properties of vertical current shear across stratification in the North Sea, Journal of marine research, 58, 465-491, 2000.
- Yanagi, T., Sachoemar, S. I., Takao, T., and Fujiwara, S.: Seasonal Variation of Stratification in the Gulf of Thailand, Journal of Oceanography, 57, 461-470, 2001.
- Zhang, N., Wang, J., Wu, Y., Wang, K.-H., Zhang, Q., Wu, S., You, Z.-J., and Ma, Y.: A modelling study of ice effect on tidal damping in the Bohai Sea, Ocean Engineering, 173, 748-760, 2019.

**Response to comments by Reviewers #2**

(1) The seasonal variation of the principal tidal constituents M2, S2, K1 and O1 is studied in this discussion paper by analyzing the sea level observations at two stations in the Bohai Bay. The authors emphasize that in the previous studies the seasonal variation of the M2 constituent has been fully investigated, while the other three have not been investigated. The authors further show that large semi-annual variation exists in S2 and K1 in their analyzed results. Since the paper contains some important improper treatments in data analysis, this paper needs major revision before publication.

**Reply:**

Thanks a lot for the positive assessment and constructive comments of our paper. The improper treatments have been modified in the revised manuscript and shown in the reply of following comments.

(2) The paper analyzes the observations monthly (month by month) to reveal seasonal variations of the obtained harmonic constants. The S2 and K1 results (Figures 3-5) show unreasonably large semi-annual (6-month period) modulation. I judge that this is due to the improper treatment in the harmonic analysis. That is, the unresolved constituents have not been removed in the analysis, resulting in spurious seasonality. To explain this, let us consider, as an example, the superposition of K1 and P1.

**• • • • • • • •**

In the above we only show the influence of P1 on K1. In fact, the constituent  $\phi_1$  may also cause (but with smaller magnitude) semi-annual modulation in K1; while  $\psi_1$ , and S1 may cause annual modulation in K1, and  $\pi_1$  may cause ter-annual (four-month) modulation in K1. Furthermore, it should be noticed that the S1 constituent actually contains two parts: the astronomical part having an amplitude ratio of 0.78% to K1; and the radiational part with magnitude depending on the strength of diurnal meteorological forcing. The latter part is

generally much greater than the former in the coastal seas, and is actually not separable from the annual variation of K1 by means of data analysis.

**Reply:**

Thank you for pointing out these improper treatments in the old manuscript and demonstrating the influences of unresolved constituents on the corresponding principal tidal constituents using the theoretical method. We have learned a lot from this comment and thanks again.

According to this comment and the fifth comment, we have redone the data analysis. (1) Consider the nodal correction and infer unresolved constituents in SHA using the function in T\_TIDE; (2) Embed the nodal correction into EHA to eliminate the influence of 18.6-year nodal modulation and add the minor constituents, whose harmonic parameters are assumed to be constant, to resolve more constituents. The details are as follows:

**(1) Selection of the unsolved constituents**

The power spectral densities of the observed sea level at E2 are shown in Figure R1 (i.e., Figure 3 in the revised manuscript), which indicated that  $P_1$  was significant, while  $S_1$  was not significant. So only  $P_1$  was considered to be the unsolved constituent near  $K_1$  and the other unsolved constituents with minor influences were ignored. Similarly, only  $K_2$  was considered to be the unsolved constituents near  $S_2$ . At tidal gauge station Dalian, which was added according to the second comment,  $P_1$  and  $K_2$  were considered to be the unsolved constituents in the diurnal and semidiurnal frequency bands, respectively, according to the power spectral densities of the sea level observations in Figure R2 (i.e., Figure 5 in the revised manuscript).

Figure R1. Power spectral densities of the observed sea level at E2 (black line) in (a) all frequency bands, (b) the diurnal frequency band, and (c) the semidiurnal frequency band. In all panels, black dashed lines denote the corresponding 5% significance level against red noise.

---

## Author Response (AR2)

Dear Editor,

We'd like to thank you for your excellent work on our manuscript numbered "os-2019-43". Based on your comments, we have modified the manuscript. Please find below a point-to-point response marked in blue, with changes in the text cited in green. The page numbers and line numbers in this document refer to the revised version of the manuscript without using the revision mode in the Microsoft Word.

(1) Reviewer 1 highlighted the necessary comparison to other published work. You now acknowledge that work, but currently you expect your readers to read all the papers and compare them. But how do your results compare to those published elsewhere? Could you for example have a table or figure quoting results from other studies and directly comparing them to yours? They may not be exactly the same sites or time periods, but please help your readers. Then, if you see the same behaviour, great! If they are different, then why? It's not really enough to say "oh yes, such-and-such a study looked at the same area before".

**Reply:**

Thank you for your constructive suggestions.

We are sorry that we did not fully compare our results to those in the published papers. According to your comment, we have further compared them. As there are spatial and temporal inhomogeneities in the seasonal variations of the principal tidal constituents and the accurate results in the published papers are difficult to extract, it is not feasible to quantitatively compare our results to the published results using a table or figure. Therefore, we have added some new comparisons in the Sections 5 (Discussion).

Compared to the annual averages, the mean $M_2$ amplitude at E2 in this study increases by 6.90 cm (approximately 9.33%) in the summer and decreases by 6.68 cm (approximately 9.03%) in the winter, close to the estimated values in Foreman et al. (1995) (6%), Huess and Andersen (2001) (6%) and Müller et al. (2014) (5%–10%). Besides, the spatial distribution of the differences between the simulated $M_2$ tidal amplitudes in summer and those in winter in Figure 10 shows that the $M_2$ tidal amplitudes in summer are larger than those in winter in Bohai Bay,

Laizhou Bay and Liaodong Bay, which is the same as that obtained by analysing the sea level data at several tidal gauge stations in the Bohai Sea in Fang and Wang (1986) and that simulated by numerical model in Kang et al. (2002). Besides, the summer amplitudes of $M_2$ tide in some areas in the Bohai Sea are less than those in winter, as shown in Figure 10, and the similar results are also obtained in Kang et al. (1995), Muller et al. (2014) and Devlin et al. (2018).

In general, the $M_2$ tidal amplitude in summer is larger than that in winter in many areas, including the Bohai Sea (Fang and Wang, 1986), the North Sea (Huess and Andersen, 2001; Gräwe et al., 2014; Müller et al., 2014), most of the Ganges-Brahmaputra-Meghna delta (Tazkia et al., 2017), the seas of Southeast Asian (Devlin et al., 2018), Liverpool (Corkan, 1934), Victoria (Foreman et al., 1995), the western part of the Yellow and East China Seas (Kang et al., 2002), the Hudson Bay and Foxe Basin (St‐Laurent et al., 2008) and so on, which is the same as that obtained by analysing the observations at E2 and Dalian in this study. Devlin et al. (2018) found that the diurnal and semidiurnal tidal amplitudes and phases exhibited a high degree of seasonality in the seas of Southeast Asia, and Fang and Wang (1986) indicated that the $O_1$ tidal amplitude in summer was also larger than that in winter in the Bohai Sea, which are similar to those concluded in this study.

It can be seen that the seasonal variations of the principal tidal constituents obtained by analysing the observations at E2 and Dalian and those simulated using numerical model in this study have the similar trend with those in the published papers. The novel EHA is used to estimate the seasonal variations and the numerical experiments using three-dimensional MITgcm are performed to explore the physical mechanisms in this study, so this study is not the simply repetitive work of any published papers.

Changes:

P9L6: For the semi-diurnal tides $M_2$ and $S_2$, the simulated amplitudes in summer were larger than those in winter in Bohai Bay, Laizhou Bay, Liaodong Bay, which was the same as that obtained by analysing the sea level data at several tidal gauge stations in the Bohai Sea in Fang and Wang (1986) and that simulated by numerical model in Kang et al. (2002), and smaller than those in winter in the middle of the Bohai Sea.

P10L6: The seasonality of the principal tidal constituents has been investigated widely. As shown in Müller et al. (2014), there were significantly seasonal variations in $M_2$ tide in several coastal regions and the maximum annual tide was in July (±1 month) in most of the ocean. However, the spatial and temporal inhomogeneities were also existed and the summer amplitudes of $M_2$ tide were less than those in winter in several areas, as shown in Kang et al. (1995), Müller et al. (2014), Devlin et al. (2018) and Figure 10 in this study. In general, the $M_2$ tidal amplitude in summer was larger than that in winter in many areas, including the Bohai Sea (Fang and Wang, 1986), the North Sea (Huess and Andersen, 2001; Gräwe et al., 2014; Müller et al., 2014), most of the Ganges-Brahmaputra-Meghna delta (Tazkia et al., 2017), the seas of Southeast Asian (Devlin et al., 2018), Liverpool (Corkan, 1934), Victoria (Foreman et al., 1995), the western part of the Yellow and East China Seas (Kang et al., 2002), the Hudson Bay and Foxe Basin (St‑Laurent et al., 2008), and so on, which was the same as that obtained by analysing the observations at E2 and Dalian in this study. Devlin et al. (2018) found that the diurnal and semidiurnal tidal amplitudes and phases exhibited a high degree of seasonality in the seas of Southeast Asia, and Fang and Wang (1986) indicated that the $O_1$ tidal amplitude in summer was also larger than that in winter in the Bohai Sea, which were similar to those concluded in this study. The seasonality of the principal tidal constituents obtained in the study was mainly similar to those in previous studies, but the novel EHA was firstly used to estimate the seasonal variations of the principal tidal constituents and the numerical experiments using three-dimensional MITgcm were performed to explore the physical mechanisms.

(2) Related to Reviewer 1 (comment 14) and Reviewer 2 (comment 2).

I'm not certain that what you are calling "M2", "S2" etc are actually the same thing when calculated by the different techniques. Each of the amplitudes A in Equations 1 and 2 are only defined as part of a whole system. As soon as you change the number of constituents fitted, and particularly whether A is time-varying, then you've changed the whole equation. I wonder if this is related to the huge N2 in the figure 5c? Can you explain this better, before I ask the reviewers to have another look?

**Reply:**

Thank you for your comment.

Tides originate from the gravitational forces of the sun and moon acting on a rotating earth (Newton, 1687). The astronomical forcing for tides can be written as a linear combination of sinusoidal terms, each having a distinct amplitude, phase lag and temporal frequency (Doodson, 1921). The oceanic response to this forcing can be expressed in the same manner, where each sinusoid is referred to as a tidal constituent (Foreman and Henry, 1989). The harmonic analysis requires calculating the amplitudes and phase lags for a finite number of sinusoidal functions with known frequencies (Foreman and Henry, 1989). In theory, when the number of constituents fitted is changed, the estimated harmonic constant of the tidal constituents will not be changed, as long as the suitable method is used.

The main points of the 14[th] comment of Reviewer #1 and the second comment of Reviewer #2 are that the estimated harmonic parameters in the old manuscript contain the other signals and the spurious seasonality is caused by the unresolved constituents. According to the fifth comment of Reviewer #2, we have used the inference procedure in the monthly analysis in SHA to remove the influence of unresolved constituents. Besides, we have embedded the nodal correction into EHA to eliminate the influence of 18.6-year nodal modulation and added the minor constituents with constant harmonic parameters in EHA to resolve more constituents in the revised manuscript. As shown in Figures 4 and 6 in the revised manuscript, the estimated seasonal variations of the principal tidal constituents using EHA are similar to those obtained using SHA; in addition, the spurious seasonality is removed, indicating that the modified methods are reasonable and the harmonic parameters of tidal constituents calculated by the different techniques are the same things.

The power spectral densities of the observed sea level at Dalian, as shown in Figure 5, are used to demonstrate the existing tidal constituents in the observations. The huge $N_2$ in Figure 5c indicates that the $N_2$ constituent is significant at Dalian. The $N_2$ in Figure 5c is the actual signal in the sea level observations and not related to the estimated harmonic parameters of the other tidal constituents. To further show that the huge $N_2$ will not affect the estimated seasonality

of the tidal constituents, we have designed new experiments NE1-NE3. In the revised manuscript, only $M_2$, $K_1$, $S_2$ and $O_1$ were selected as the principal tidal constituents to be investigated using SHA and EHA. In SHA, the automated constituent selection algorithm in T_TIDE was used to determine the analysed constituents ($N_2$ was included in fact); in addition, the unresolved constituents $P_1$ and $K_2$ were inferred from $K_1$ and $S_2$, respectively, with the inference parameters taken from a yearly harmonic analysis of the sea-level observations. In EHA, the harmonic parameters of $M_2$, $K_1$, $S_2$, $O_1$, $P_1$ and $K_2$ were estimated together; in addition, the harmonic parameters of $M_2$, $K_1$, $S_2$ and $O_1$ were assumed to be temporally varying and those of $P_1$ and $K_2$ were assumed to be constant. In new experiment NE1, the $N_2$ was added to be principal tidal constituents with temporally varying harmonic parameters and estimated together with the other tidal constituents using EHA. The other settings of EHA in NE1 were the same as those in the EHA in the revised manuscript. As shown in the following Figure R1, the estimated harmonic parameters of $M_2$, $K_1$, $S_2$ and $O_1$ were almost equal to those in the revised manuscript. In new experiment NE2, the $N_2$ was assumed to be the minor tidal constituents with constant harmonic parameters and estimated together with the other tidal constituents using EHA. The other settings of EHA in NE2 were the same as those in the EHA in the revised manuscript. As shown in the following Figure R2, the estimated harmonic parameters of $M_2$, $K_1$, $S_2$ and $O_1$ in NE2 were almost equal to those in the revised manuscript. In new experiment NE3, only $M_2$, $S_2$, $K_1$ and $O_1$ were estimated together using SHA and the unresolved constituents $P_1$ and $K_2$ were inferred from $K_1$ and $S_2$; in fact, the $N_2$ was not considered. The other settings of SHA in NE3 were the same as those in the SHA in the revised manuscript. As shown in the following Figure R3, the estimated harmonic parameters of $M_2$, $K_1$, $S_2$ and $O_1$ were almost equal to those in the revised manuscript. The results indicated that the estimated harmonic parameters of the principal tidal constituents in the revised manuscript are not changed, no matter the $N_2$ is considered or not in both EHA and SHA.

[Figure]

Figure R1. Time series of the estimated temporally varying tidal amplitudes of principal tidal constituents, including (a) $M_2$, (b) $S_2$, (c) $K_1$ and (d) $O_1$, at Dalian using EHA in the revised manuscript (red lines) and in the new experiment NE1 (blue lines). (e-h) Similar to (a-d), but for the estimated temporally varying tidal phase lags. Shading indicate the corresponding 95% confidence intervals.

[Figure]

Figure R2. Similar to Figure R1, but for those at Dalian estimated using EHA in the revised manuscript (red lines) and in the new experiment NE2 (blue lines).

[Figure]

Figure R3. Time series of the estimated temporally varying tidal amplitudes of principal tidal constituents, including (a) $M_2$, (b) $S_2$, (c) $K_1$ and (d) $O_1$, at Dalian using SHA in the revised manuscript (blue lines) and in the new experiment NE3 (red lines). (e-h) Similar to (a-d), but for the estimated temporally varying tidal phase lags. Vertical bars indicate the corresponding 95% confidence intervals.

[revised manuscript text omitted]

---

## Author Response (AR3)

Dear Editors,

Thank you for your excellent work on our manuscript. We'd like to thank the reviewers for their comments and positive appraisal. Based on their comments, we have changed the manuscript. Please find below a point-to-point response marked in blue, with changes in the text cited in green. The page numbers and line numbers in this document refer to the revised version of the manuscript without using the revision mode in the Microsoft Word.

**Response to comments by Reviewer #1**

(1) -OVERVIEW: -This is my second review of this paper, which analyzes the seasonal variation of tidal constituents in the Bohai Sea, using data from a mooring as well as tide gauge data from Dalian. Additionally, the study includes some modelling work to help explain the observed seasonality. I believe that this work is important and interesting, as the nonastronomical seasonality of tides is an important topic. This effect has been examined in many locations, but to my knowledge, no other studies have looked specifically at the Bohai Sea, so this work is a welcome addition to the body of knowledge. This is especially true for this region of China, where there is not much publicly available tide gauge data to utilize. Furthermore, I think that the authors have done a good job of explaining the past efforts of previous studies, discussed well the possible mechanisms behind the observed seasonality, and have also performed some well-designed modelling experiments to back up their observations. Finally, this version is much improved from the first submission, and the majority of my initial concerns have been adequately covered, so I am very pleased with the current form of the paper, and believe it should be an important submission to help guide future investigations of tidal seasonality. My recommendation for this paper is for acceptance, though I do have a small number of (mostly technical) comments and thoughts that I would like to see addressed first.

**Reply:**

Thanks a lot for the positive assessment of the manuscript. Your constructive comments in the first review gave us many suggestions to further improve the old manuscript, thank you again. (2) -In the abstract, you talk about the location of your mooring observations, which you give as "E2". However, at this stage of the paper, the reader does not know what "E2" means, since you have not yet told us that this is the name given to your mooring location; this is described later. Therefore, in the abstract, you should instead just mention the approximate geographical location of the mooring observations, which should be something like "the western Bohai Sea", or something similar. "E2" is just a shorthand indicator to tell us about a real location, so be clear about this in the abstract, and after you describe the exact location of "E2", then you can rely on using the shorthand.

**Reply:**

Thank you for your constructive comment. According to your suggestion, we have added the description of the mooring station when the "E2" firstly appeared, as shown in following 'Changes'.

Changes:

P1L11-P1L14: In this study, seasonal variations of four principal tidal constituents, M2, S2, K1, and O1, in the Bohai Sea, China, were studied by applying the enhanced harmonic analysis method to two time series: one-year sea level observations at a mooring station (named E2) located in the western Bohai Sea and 17-year sea level observations at Dalian.

(3) -I think that is good that you compare both the mooring data, which is a short record of about a year from 2013-2014, and the tide gauge data from Dalian/Laohutan from 1980-1997. As I have also tried working with the tide gauge data from Mainland China, and understand that there is not much recent publicly available data, the "historical" tide gauge records that all end in the 1997 are the only resources available to employ. However, I think that you should include some caveats in your discussion about the fact that these two sets of data come from different time eras, and that there might be a limit to have comparable the two data sets might be. In particular, there has been some previous suggestions that some major shifts in climate and sea level behavior occurred around 1998, roughly coinciding with the 1997-1998 El Nino event, and this "shift" may have changed many of the physical oceanic properties of the Western

Pacific, especially things like sea level rates and upper-ocean warming, which both tended to increase quite a bit in the post-1998 era. I still think that your present comparison is valid, I am just saying that some mention about the time difference should be included.

**Reply:**

Thank you for your constructive comments. As you said, the publicly available sea level observations in the Bohai Sea are only at Dalian and there are indeed no other publicly available observations. The 1997-1998 El Nino is one of the strongest in the twentieth century (Chavez et al., 1999), which will indeed affect the physical oceanic properties. According to your suggestions, we have added the discussion about the time difference in Section 5, as shown in the following 'Changes'.

**Changes:**

P9L31-P10L2: It is noted that the duration of sea level observations at E2 is from 2013 to 2014 and that for Dalian is from 1980 to 1997, which are from different eras and may defy comparison because the 1997-1998 El Nino is one of the strongest in the twentieth century (Chavez et al., 1999) and changes many of the physical oceanic properties (Nezlin and Mcwilliams, 2003; Shang et al., 2005; Liu et al., 2010).

(4) -Please give a little more description about the combination of data from the Dalian and Laohutan data sets; saying they "shared position" does not seem to make sense, as these are two separate locations. Do you mean to say that the gauge was "relocated", but the two locations are "close enough" to consider them to be combineable? UHSLC gives some history of the gauges and history in the metadata links of their data archives, and you should also be able to find some helpful information about each gauge from the PSMSL data overviews. For example, is there any reasons given as to why the gauges were moved, and did anyone validate their connections?

**Reply:**

Thank you for your constructive suggestions. We are sorry that the description of the combination of data from Dalian and Laohutan was insufficient in the old manuscript.

As shown in the information of hourly UHSLC Tide Gauge Data ("Research Quality") (http://uhslc.soest.hawaii.edu/thredds/uhslc\_quality\_hourly.html), the longitude, latitude and duration of the sea level observations at Dalian are 38.933°N, 121.667°E and 1975-1990, respectively, while those for Laohutan are 38.867°N, 121.683°E and 1991-1997, respectively. Furthermore, Feng et al. (2015) pointed out: "The nearby Dalian and Laohutan stations have data for the periods 1975–1990 and 1991–1997, respectively. However, at Dalian the tide gauge was relocated twice, once at the end of 1976, and once in 1979. After 1979, it shared the same position as Laohutan. Thus, the data for the periods 1980–1990 at Dalian and 1991–1997 at Laohutan were combined.". Therefore, the Dalian station was relocated in 1979 and these two locations are close enough to consider them to be combinable. According to your suggestions, we have added the detailed descriptions of the data combination, following Feng et al. (2015). Changes:

P2L30-P3L2: As indicated in Feng et al. (2015), the nearby Dalian and Laohutan stations have data for the period 1975-1990 and 1991-1997, respectively; in addition, the Dalian station was relocated in 1976, and again in 1979, after which Dalian station shared the same position as Laohutan. Therefore, following Feng et al. (2015), the tidal gauge station data at Dalian used in this study were comprised of data from Dalian from 1980-1990 and that from Laohutan from 1991-1997, as shown in Figure 2b.

(5) -In Section 4, where you discuss possible mechanisms, you might consider adding monsoonrelated mechanisms as another possibility. Wind forcing changes, such as Ekman forcing and changes in geostrophic in forcing was shown to be possibly important by Devlin et al, 2018 in explaining the seasonality of the tides in SE Asia. Others might have found similar conclusions. **Reply:**

Thank you for your constructive suggestion. According to your suggestion, we have added the monsoon to be the possible mechanisms in the revised manuscript, as shown in the following 'Changes'.

In addition, we have found an important review paper (Talke and Jay, 2020, Annual

Review of Marine Science), in which the role of natural and anthropogenic factors in changing tides is reviewed. This review paper is first posted online on September 3, 2019 and the expected final online publication data is January 3, 2020. It has been added in the revised manuscript. As you have mentioned that you have also tried working with the tide gauge data, we want to share this review paper with you.

**Changes:**

P7L4-P7L8: Other mechanisms, including long-term changes in the tidal potential (Molinas and Yang, 1986), monsoon (Devlin et al., 2018), interactions with other physical phenomena (Huess and Andersen, 2001; Pan et al., 2018a), changes in the internal tide with corresponding small changes in its surface expression (Ray and Mitchum, 1997; Colosi and Munk, 2006), as well as a number of technical reasons, may also change the M2 amplitude on various time scales. The above reasons have been presented or discussed in Woodworth (2010), Müller (2012), Müller et al. (2014), Tazkia et al. (2017), and Talke and Jay (2020).

(6) -A quick comment about section structure. Section 3 is labelled "Results", but Section 4.2 is also labelled "Results", which is a bit confusing. I think that maybe the latter instance refers to "Modelling Results", so this might be a better title for this section.

**Reply:**

Thank you for your constructive suggestion. According to your suggestion, we have modified it in the revised manuscript, as shown in the following 'Changes'.

Changes:

**P8L6: 4.2 Modelling Results**

(7) -A final comment is just about the English grammar. This version is markedly improved from the first version, however, there are a small number of relatively minor grammar issues. It is probably OK as-is, but my gentle recommendation is that the author could do well by giving the grammar one more scan, and perhaps ask the help of a native English speaker or editing service to give it a close look. Cleaning up some of the minor issues with the grammar would improve this good paper even more.

**Reply:**

Thank you for pointing this out. We are sorry that the old manuscript polished by the language editor was still not good. When we modified this manuscript, Zheng Guo, who was an editor in an English journal of oceanography, was invited to further polish our manuscript.

**References**

- Chavez, F. P., Strutton, P. G., Friederich, G. E., Feely, R. A., Feldman, G. C., Foley, D. G., and Mcphaden, M. J.: Biological and Chemical Response of the Equatorial Pacific Ocean to the 1997-98 El Niño, Science, 286, 2126-2131, 1999.
- Devlin, A. T., Zaron, E. D., Jay, D. A., Talke, S. A., and Pan, J.: Seasonality of Tides in Southeast Asian Waters, Journal of physical oceanography, 48, 1169-1190, 2018
- Feng, X., Tsimplis, M. N., and Woodworth, P. L.: Nodal variations and long term changes in the main tides on the coasts of C hina, Journal of Geophysical Research: Oceans, 120, 1215-1232, 2015.
- Liu, X., Liu, Y., Lin, G., Rong, Z., Gu, Y., and Liu, Y.: Interannual changes of sea level in the two regions of East China Sea and different responses to ENSO, Global & Planetary Change, 72, 215-226, 2010.
- Nezlin, N. P. and Mcwilliams, J. C.: Satellite data, Empirical Orthogonal Functions, and the 1997–1998 El Niño off California, Remote Sensing of Environment, 84, 234-254, 2003.
- Shang, Zhang, S., Hong, C., Liu, H. S., Wong, Q., Hu, G. T. F., and Huang, C.: Hydrographic and biological changes in the Taiwan Strait during the 1997–1998 El Niño winter, Geophysical Research Letters, 32, 343-357, 2005.
- Talke, S. A. and Jay, D. A.: Changing Tides: The Role of Natural and Anthropogenic Factors, Annual review of marine science, 12, 14.11-14.31, 2020.

**Response to comments by Reviewer #2**

(1) This revised manuscript has eliminated the major problems contained in the original one, and thus can be published after minor revision given below.

**Reply:**

Thanks a lot for the constructive comments and the positive assessment of the manuscript. According to your comments, we have modified the manuscript. The details are shown in the replies of the following comments.

(2) Page 2, lines 24-25: Change "Hourly sea-level data at the Dalian tidal gauge station were obtained from the University of Hawaii Sea Level Center and used. After 1979, Dalian shared position with Laohutan (Feng et al., 2015)" to "Hourly sea-level data at the Dalian tidal gauge station used in this study were obtained from the University of Hawaii Sea Level Center. After 1979, the Dalian station is located at Laohutan (Feng et al., 2015)".

**Reply:**

Thank you for pointing this out. In the revised manuscript, we have changed the corresponding sentence and rewritten this paragraph, as shown in the following 'Changes'. Changes:

P2L29-P3L2: Hourly sea level data at the Dalian tidal gauge station used in this study were obtained from the University of Hawaii Sea Level Center. As indicated in Feng et al. (2015), the nearby Dalian and Laohutan stations have data for the period 1975-1990 and 1991-1997, respectively; in addition, the Dalian station was relocated in 1976, and again in 1979, after which Dalian station shared the same position as Laohutan. Therefore, following Feng et al. (2015), the tidal gauge station data at Dalian used in this study were comprised of data from Dalian from 1980-1990 and that from Laohutan from 1991-1997, as shown in Figure 2b

(3) Page 4, line 7: Change "oscillations" to "fluctuations".

**Reply:**

Thank you for pointing this out. In the revised manuscript, we have changed it, as shown in P4L11.

(4) Page 4, line 23: Delete "and near those estimated using CHA", because the parameters derived from CHA do not have temporal variation, see Figure 4.

**Reply:**

Thank you for your suggestion. We want to say that the averaged values of the harmonic parameters estimated using SHA and EHA are near to that estimated using CHA, which shows that the estimated results using SHA and EHA are reasonable. Therefore, we did not delete it and modified the sentence, as shown in the following 'Changes'. Changes:

P4L26-P4L29: As shown in Figure 4, the estimated harmonic parameters obtained with SHA and EHA, including the temporally varying amplitudes and phase lags, were nearly equal and the averaged values were near to that estimated using CHA, indicating that the temporal variations in the harmonic parameters of the principal tidal constituents at E2 can be reasonably estimated using both SHA and EHA.

(5) Page 5, lines 20-23: Change "The estimated harmonic parameters using both the SHA and EHA were near to those obtained using CHA, showing that the estimated results were reasonable. In addition, the estimated harmonic parameters obtained using EHA were much closer to those obtained using SHA for data from Dalian than those from E2" to "The averaged harmonic parameters estimated using both the SHA and EHA were close to those obtained using CHA, showing that the estimated results were reasonable. In addition, the estimated are using both the SHA and EHA were close to those obtained using CHA, showing that the estimated results were reasonable. In addition, the estimated harmonic parameters obtained using EHA were much closer to those obtained using SHA for data at Dalian than those at E2".

**Reply:**

Thank you for pointing it out. According to your suggestion, we have changed the corresponding sentence in the revised manuscript, as shown in P5L22-P5L25.

(6) Page 7, lines 17-18: Change "The simulation area was the Bohai Sea as shown in Figure 1b" to "The simulation area of the Bohai Sea is shown in Figure 1b".

**Reply:**

Thank you for pointing it out. According to your suggestion, we have changed the corresponding sentence in the revised manuscript, as shown in P7L18-P7L19.

(7) Page 12, line 11: Change "Weiwen" to "Wenzheng"; change "Zhonzhu" to "Zongyong".Reply:

Thank you for pointing this out. We are sorry that the author names in this literature were wrong, because we managed the literatures using ENDNOTE and there were errors in the downloaded ENDNOTE file of this literature. In the revised manuscript, we have modified this literature, as shown in P12L24.

(8) Page 12, line 45: Replace "Accepted, 2018" with exact volume and page numbers, because this paper should have already been printed.

**Reply:**

Thank you for pointing this out. We are sorry that we did not update this literature in the old manuscript. According to this comment, we have modified this literature in the revised manuscript, as shown in P13L16.

(9) Page 14, Tables 1 and 2: Change all "phase" to "phase lag", because in the literatures on ocean tides, "phase lag" is convectional used instead of "phase".

**Reply:**

Thank you for pointing these out. We are sorry that we did not modify them in the Tables 1 and 2 in the old manuscript. According to this comment, we have modified them in the revised manuscript.

**Response to comments by Reviewer #3**

(1) I wonder why the authors use only E2 and Dalian data in this paper. In the Bahai Sea, there are other tide gauges such as QinghuangDao and Tanggu. It would be more convincing by analyzing ALL available data sets for exploring seasonal variability. In addition, I am not sure the SSH data at a cross-over point of satellite altimeter tracks in the central Bohai Sea is long enough for the seasonal study. At present, E2 and Dalian data are studied, but they have different lengths and qualities. Why not conduct a COMPLETE investigation using all these data sets? **Reply:**

Thanks a lot for the constructive comments of our paper. Although there are many tidal gauge stations in the Bohai Sea, the publicly available sea level observations are scarce. We can only obtain the hourly sea level observations with research quality at Dalian from University of Hawaii Sea Level Center (UHSLC). Because of the data privacy policy on the sea level observations at tide gauge stations, the authors, who have published papers using the data at the other tidal gauge stations in the Bohai Sea, cannot share the data with us. We are sorry that we do not have ways to obtain the hourly sea level observations at other tidal gauge stations in the Bohai Sea.

As only the regularly sampled observations can be analyzed by the current version of enhanced harmonic analysis used in this study, the SSH data at a cross-over point of satellite altimeter tracks is not considered. In addition, as indicated by Muller et al. (2014), estimating the seasonal cycle from satellite altimetry data is possible, but limited to a few regions where the signal is large enough to exceed the background noise level. As mentioned in Shum et al. (1997) and Fang et al. (2004), the harmonic parameters derived from TOPEX/Poseidon alongtrack altimetry in the shallow area (e.g., the Bohai Sea) are not as accurate as in deep oceans, so if the SSH data in TOPEX/Poseidon can be used to study the seasonal variations of tides in the Bohai Sea is still a problem and beyond the scope of this paper. Thank you for your suggestions again. We will try to investigate the seasonality of tides in the China Seas using satellite data in the future.

Although the hourly sea level observations at E2 and Dalian have different lengths and

qualities, the seasonality of the principal tidal constituents at E2 and Dalian can be analyzed. We agree that there might be a limit to compare the estimated results. As suggested by Reviewer #1, we have added the discussion about the time difference in Section 5.

As described above, we have used all the available hourly sea level observations (within our reach) to investigate the seasonal variations in the principle tidal constituents in the Bohai Bay. The limitation of time difference of the observations used in this study is discussed in the Section 5 (Discussions), as shown in the following 'Changes' Changes:

P9L31-P10L6: It is noted that the duration of sea level observations at E2 is from 2013 to 2014 and that for Dalian is from 1980 to 1997, which are from different eras and may defy comparison because the 1997-1998 El Nino is one of the strongest in the twentieth century (Chavez et al., 1999) and changes many of the physical oceanic properties (Nezlin and Mcwilliams, 2003; Shang et al., 2005; Liu et al., 2010). In addition, the duration of hourly sea level observations at E2 used in this study is only one year, which is also a limitation. Although the different time eras and the rare event might slightly skew seasonal pattern of tides, the seasonal variations of the principal tidal constituents M2, S2, K1 and O1 obtained using EHA are the same as those using traditional SHA, indicating that the results are not unreasonable and can reflect the seasonal variations of tides in the analysis period.

(2) I found the four numerical experiments are problematic. Look at Figure 9, one cannot draw any solid conclusions about the effect of stratification and vertical viscosity coefficient. The numerical and observed results do not agree with each other in most cases. Thus, the authors' results, as described in section 4.2, are suggestive but not conclusive. The authors better design experiments that may lead to "conclusive" results.

**Reply:**

Thank you for your constructive comments.

It is true that the simulated and observed results do not agree with each other in most cases, as many other factors influencing the simulated results are not considered in the numerical experiments. For example, the spatially and temporally varying ocean circulation, the sea ice appeared in winter in the Bohai Sea, the spatially varying bottom drag coefficient, the accurate water depth and other factors are not included in the numerical models. Muller et al (2014) used an ocean circulation and tide model with some advanced parameterization schemes and model settings to investigate the seasonal variation of the M2 tide globally. In their study, for the East China and Yellow Seas, only in the western Yellow Sea the simulated results at tidal gauge stations were consistent with the observed seasonality of the M2 tide, whose maxima in summer months were about 0.02-0.04 m, while the seasonality between Taiwan and China and also at tidal gauge stations along the South Korean coast were not well captured in the simulated seasonal cycle. Therefore, it is not realistic to make the simulated results be very close to the observed values for every tide at both E2 and Dalian.

As described in the manuscript, the numerical experiments were used to test the influence of seasonally varying possible factors on the seasonal variability of the principal tidal constituents. Therefore, the variation trends of the simulated harmonic parameters of the principal constituents are used to test the possible mechanisms. The simulated M2 amplitude at E2 in winter is almost equal to the observed value in winter, in addition, the simulated M2 amplitude at E2 in Exp3 is also nearly equal to the observed value in summer, indicating that the numerical model has certain ability to simulate the tides in the Bohai Sea. From the changes between the simulated results in summer (Exp2, Exp3 and Exp4) and those in winter, it is obvious that the seasonal variation trends of the tides in the observations are captured by the Exp3, except  $K_1$  phase lag. Therefore, seasonal variation in the vertical eddy viscosity is the most important mechanism influencing the seasonal variability of principal tidal constituents at E2. At Dalian, the simulated harmonic parameters of the principal constituents in the numerical experiments are not consistent with the observed values, but the differences between the simulated results in winter and those in summer with changing stratification and vertical eddy viscosity have the same trends with the observed differences in much more cases than other experiments, so the seasonal variations of the principal tidal constituents at Dalian are possibly determined by the seasonality of stratification and vertical eddy viscosity

We have tried our best to design and run much more numerical experiments, including increasing the simulation area, using water depth from other sources, and so on, but the simulated results were not better than those shown in the manuscript to further explain the seasonal variations of the tides in the Bohai Sea. Although the conclusions from the numerical results are suggestive, it can help us to understand the mechanisms of the observed seasonality of the tides in the Bohai Sea to some extent. As you said, the "conclusive" results are the best, which is also our aim of struggle. As you mentioned in the comments, we will try to investigate the seasonality of tides in the China Seas using satellite data and further improve the numerical model in the future.

(3) The citations are confusing. The authors do cite a lot of papers on this topic. But they just CITE them, not really digest them. For example, some results about seasonal variability are based on modeling work, with which the authors better be alert (e.g. Muller et al. 2014). Another example, P7 L76, this sentence, citing 4 papers, means nothing without details. Another example, P6 L30 says that seasonal ice coverage is a significant reason; while P7 L11 says that Zhang et al (2019) has rejected this effect.

**Reply:**

Thank you for your constructive comments.

It is indeed that the seasonal variations of the M2 tide estimated in Muller et al. (2014) and some other papers are based on modeling work, but we should notice that traditionally their models are firstly evaluated. In addition, the modeling results are analyzed on a large spatial scale, which is difficult to realize using the sea level observations, as the tidal gauge stations are just in the coastal ocean and the estimation of seasonal cycle from satellite altimetry data is limited to a few regions. Therefore, the modeling results of the seasonal variations in the tides are referred. Of course, we have firstly evaluated the results and conclusions based on our knowledge.

In P7L7 of the old manuscript, we want to say that the mentioned possible mechanisms in this part are also summarized in those four literatures in other ways. We think that we should cite them and help the readers to easily find the other papers in which the seasonal variations of the tides and the corresponding possible mechanisms are investigated and discussed.

In P6L30 of the old manuscript, we want to say that the seasonally varying ice coverage is possible main mechanisms of the seasonal variations in tides. It should be noted that this main mechanism is found in the Arctic (St-Laurent et al., 2008; Georgas, 2012; Muller et al, 2014). As the Bohai Sea also freezes in some areas, this possible mechanism is listed in the manuscript. However, when we modified the manuscript based on the reviewers' comments in first review, we found the studies of Zhang et al. (2019). Zhang et al. (2019) found that the damping effect of sea ice on the astronomical tides could be almost negligible in the Bohai Sea. As we know, the physical phenomenon and mechanism have the corresponding precondition and spatial-temporal limitation, so the disagreement between the conclusion in Arctic and that in Bohai Sea is not surprising.

(4) Throughout the paper, the tide names M2, S2, O1 and K1 are in italic font. They should be in normal font as  $M{sub{2}}$  and  $S{sub{2}}$ .

**Reply:**

Thank you for pointing these out. We are sorry that the expression of the tidal names was wrong in the old manuscript. The normal font (e.g.,  $M_2$ ) was used in some literatures (e.g., St-Laurent et al., 2008; Devlin et al., 2018) and the italic font (e.g.,  $M_2$ ) was also used in some literatures (e.g., Muller et al., 2014; Talke and Jay, 2020), so we have been confused. Thank you for telling us the right expression. According to your suggestion, we have modified them in the text, Table 1, Table 2, Figure 3, Figure 5 and Figure 9 in the revised manuscript. Please see the details in the revised manuscript.

(5) The writing is still not good. Your "language editor" can make sure your English grammar is correct. But he/she cannot guarantee that your writing is accurate, concise and conniving. It is beyond their capability.

**Reply:**

Thank you for your constructive comment. We are sorry that the manuscript polished by the language editor was still not good. When we modified this manuscript, Zheng Guo, who was an editor in an English journal of oceanography, was invited to further polish our manuscript.

(6) Please give us the longitude and latitude of E2.

**Reply:**

Thank you for your comment. E2 is located at 38.65°N, 118.28°E. According to your suggestion, we have added the longitude and latitude of E2 in the revised manuscript, as shown in the following 'Changes'.

**Changes:**

P2L25-P2L27: From 0000 UTC 1 November 2013 to 0000 UTC 1 November 2014, total sea levels were observed hourly using a moored pressure gauge accurate to within 5 cm (Lv et al., 2019), at E2 station (38.65°N, 118.28°E) in the Bohai Bay, China (Figure 1).

(7) Some "scientific facts" better be in present tense, not past tense.

**Reply:**

Thank you for pointing these out. We are sorry that the improper tense was used in the old manuscript. According to your suggestion, we have modified them in the revised manuscript, and some examples are shown in the following 'Changes'.

**Changes:**

P1L11-P1L21: The seasonal variation of tides plays a significant role in water level changes in coastal regions. In this study, seasonal variations of four principal tidal constituents, M2, S2, K1, and O1, in the Bohai Sea, China, were studied by applying the enhanced harmonic

analysis method to two time series: one-year sea level observations at a mooring station (named E2) located in the western Bohai Sea and 17-year sea level observations at Dalian. At E2, the  $M_2$  amplitude and phase lag have annual frequencies, with large values in summer and small values in winter, while the frequencies of  $S_2$  and  $K_1$  amplitudes are also nearly annual. In contrast, the  $O_1$  amplitude increases constantly from winter to autumn. The maxima of phase lags appear twice in one year for  $S_2$ ,  $K_1$  and  $O_1$ , taking place near winter and summer. The seasonal variation trends estimated by the enhanced harmonic analysis at Dalian are different from those at E2, except for the  $M_2$  phase lag. The  $M_2$  and  $S_2$  amplitudes show semi-annual and annual cycles, respectively, which are relatively significant at Dalian. The results of numerical experiments indicate that the seasonality of vertical eddy viscosity induces seasonal variations of the principal tidal constituents at E2, while the variations at Dalian are possibly caused by the seasonality of stratification and vertical eddy viscosity.

(8) P1 L14, "S2 and K1 tidal amplitudes" where 'tidal' can be omitted. Many such cases in this paper.

**Reply:**

Thank you for pointing these out. According to your suggestions, we have deleted 'tidal' in the revised manuscript.

(9) In Introduction, Section 1, please start with the history and basic conclusions of the tide in the Bohai Sea, and lead to your motivating questions for this study.

**Reply:**

Thank you for your constructive comment. According to your suggestions, we have tried to describe the basic knowledge of the tides in the Bohai Sea and the motivation of this study in the revised manuscript, as shown in the following 'Changes'.

**Changes:**

P1L23-P1L28: Tidal motion is one of the major dynamical processes in the Bohai Sea and has been widely studied (Fang et al., 2004). Although there is no primary seasonal cycle in the

moon's orbit, a significant seasonal variation in the principal lunar tidal constituent has been observed and is dominant in coastal and polar regions (Müller et al., 2014). The seasonal variations of several semidiurnal tides are also found to be significant in the Bohai Sea (Fang et al., 1986), but the corresponding comprehensive investigations are sparse, which results in that the seasonal variation characteristics and mechanisms need to be further studied.

(10) Since the major topic is the seasonality of tides in the Bohai Sea, please downscale the "Methods" part. Now your comparisons of CHA, TTIDE, UTIDE, EHA are overweighted. You authors have published 2 papers on this topic in 2018.

**Reply:**

Thank you for your comment. As you can see, the comparisons in Section 5 (Discussions) were added when we modified the manuscript according to the comments of Reviewer #1 in the first review.

The enhanced harmonic analysis (EHA) is developed in Jin et al. (2018) and Pan et al. (2018), but the EHA is further improved in this study to estimate the seasonal variations in the tides. The EHA is a novel method and firstly used to investigate the seasonality of the tides, so in our opinion, it is necessary to compare it with the other methods that are developed based on T\_TIDE and have been widely used. Therefore, although the comparisons may be overweighted, we think it is not necessary to delete them in the manuscript.

(11) P9 L8, Do Fang and Wang (1986) have any other data sets or results that you can cite and compare here?

**Reply:**

Thank you for your constructive comments. Although Fang and Wang (1986) investigated the seasonality of the tides in the Bohai Sea using the sea level observations at Yingkou, Huludao, Qinhuangdao, Tanggu, Longkou, Dalian and Yantai, the sea level observations are not publicly available and we cannot obtain them. As the sea level observations used in Fang and Wang (1986) are located in many areas of the Bohai Sea, the simulated seasonal variation trends in the three bays of the Bohai Sea in our studies are compared with analyzed results in Fang and Wang (1986). In addition, Fang and Wang (1986) indicated that the  $O_1$  amplitude in summer was larger than that in winter in the Bohai Sea, which were similar to those concluded in this study and has been mentioned in P10L21 of the revised manuscript. So, the main results of the seasonal variations of the tides in the Bohai Sea in Fang and Wang (1986) have been cited and compared in this manuscript.

**(12) P10 L23, well-mixed.**

**Reply:**

Thank you for pointing this out. We have modified it in the revised manuscript.

(13) P10 L29, "sea-level" throughout the paper, such a hyphen is not necessary.

**Reply:**

Thank you for pointing these out. We have changed all of the "sea-level" to "sea level" in the revised manuscript.

(14) P11 L22, why does Shenzhen support a research in the Bohai Sea?

**Reply:**

Thank you for your comment. In fact, the scientific research projects can not only support the investigations in the special areas of the ocean near special city. Our Shenzhen project supports us to do the basic research in physical oceanography, mainly in the study of storm surge and tides. Based on the support of Shenzhen project, we have improved EHA and tried to investigate the seasonal variations in the tides. As we have the sea level observations in the Bohai Sea, the improved EHA is applied in the Bohai Sea and this paper is written. (15) Figure 2a, is the drop around 2013-12-1 caused by a winter storm? say it.

**Reply:**

Thank you for your constructive comment.

According to your comment, we download the 10 m wind in the NCEP Climate Forecast 2 Selected System Version (CFSv2) Hourly **Time-Series** Products (https://rda.ucar.edu/datasets/ds094.1/index.html#!description), as we do not have the in-situ wind observations. As shown in Figure R3-1, the 10 m wind speeds at E2 are indeed large in some cases, especially at 1200 UTC 6 November 2013 and 2100 UTC 26 November 2013. In addition, the wind speeds in most of the Bohai Sea at those times are larger than 15 m/s, which results in the sea level drops in November 2013. As this part is not the main content of the manuscript, the Figure R3-1 is not added into the revised manuscript. However, we have added the description of the sea level drops in November 2013 into the revised manuscript, as shown in the following 'Changes'.

Figure R3-1. Time series of the 10 m wind speed (blue line) and sea level (red line) at E2, and the spatial distributions of 10 m wind speed (colors) and wind direction (white arrows) at the correspond time.

Changes:

P2L28: The obvious sea level drops in November 2013 at E2 are mainly caused by the winter storms.

(16) Figure 3 and 5, can you plot two spectra, one for summer and one for winter? That is, can you see seasonal difference in spectra?

**Reply:**

Thank you for your constructive comment.

According to your suggestions, we have drawn the power spectral densities of the sea level observations in summer (June 2014 to August 2014) and winter (December 2013 to February 2014) at E2 (Figure R3-2) and Dalian (Figure R3-3). It is noted that the definitions of summer and winter are the same as those in the manuscript. As shown in Figure R3-2, the power spectral densities of  $K_1$ ,  $M_2$  and  $S_2$  in winter are smaller than those in summer, respectively, which is the same as the seasonal difference obtained using EHA in the revised manuscript (Table 1). As shown in Figure R3-3, the power spectral densities of  $K_1$  and  $M_2$  in winter are smaller than those in summer, while the power spectral density of  $S_2$  in winter is larger than that in summer, which are the same as the seasonal difference obtained using EHA in the revised manuscript (Table 2). The seasonal difference obtained from power spectral density of  $O_1$  is different from that obtained using EHA at E2, which may be because the power spectral density of  $O_1$  at E2 has obvious red shift and blue shift in summer and winter, respectively. It is noted that in the frequency bands near  $K_1$  ( $M_2$ ), the power spectral densities in both summer and winter have red (blue) shift, so the seasonal differences are the same as those obtained using EHA.

As the seasonal differences of tides obtained using harmonic analysis and power spectral analysis are almost the same and we want to show the effectiveness of EHA in estimating the seasonal variations in tides in the Bohai Sea, the results of the power spectral analysis are not shown in the revised manuscript.

---

## Author Response (AR4)

Dear Prof. Williams,

Thank you for your excellent work on our manuscript. We'd like to thank you for your comments. Based on your comments, we have changed the manuscript. Please find below a point-to-point response marked in blue, with changes in the text cited in green. The page numbers and line numbers in this document refer to the revised version of the manuscript without using the revision mode in the Microsoft Word.

(1) Thank-you for your recent revision (os-2019-43-manuscript-version6) and response to reviewers. The paper is considerably better than it started and is much clearer to read. I am satisfied in most respects that you have addressed the reviewers comments. However these still remain, and the first is a very important point.

**Reply:**

Thanks a lot for the positive assessment of the manuscript and your excellent work on our manuscript. According to your comments, we have tried our best to modify the manuscript, as shown in the following responses.

**(2) Reviewer 3, point 2:**

"2) I found the four numerical experiments are problematic. Look at Figure 9, one cannot draw any solid conclusions about the effect of stratification and vertical viscosity coefficient. The numerical and observed results do not agree with each other in most cases. Thus, the authors' results, as described in section 4.2, are suggestive but not conclusive. The authors better design experiments that may lead to "conclusive" results."

You have not addressed this sufficiently. I agree with the reviewer that the evidence presented at Dalian is really very weak, since the numerical experiments and observations are in very poor agreement, as shown in figures 9 and 10 and section 4.2. There is some evidence at E2 that favours Experiment 3.

The conclusion in P11 line 17, and the last line of the abstract are therefore not justified by the evidence you have presented. It's OK to present a negative result, that you have tested various mechanisms and have not found any that reproduce the observations. But it must be made clearer in the discussion and also in the abstract. For example in the abstract:

"The results of numerical experiments indicate that the seasonality of vertical eddy viscosity induces seasonal variations of the principal tidal constituents at E2. However these mechanisms do not reproduce the observations at Dalian."

**Reply:**

Thank you for your constructive suggestions. It is indeed obvious that the results of numerical experiments dose not reproduce the seasonality of tides at Dalian. Although we have tried to compare the simulated results with the observed seasonality, it is not good, as you said. Thank you for your comments, which make us relaxed. We agree with your suggestions and present the negative result in the revised manuscript, as shown in the following 'Changes'. Changes:

P1L21-P1L22: However, the tested mechanisms, including seasonally varying stratification, vertical eddy viscosity and mean sea level, do not adequately explain the observed seasonal variations of tidal constituents at Dalian.

P9L3-P9L5: On the whole, the seasonal variations of the principal tidal constituents at Dalian were not adequately reproduced by the numerical experiments, indicating that all the tested mechanisms were not the possible mechanism.

P10L14-P10L16: In addition, the constant harmonic parameters were used to predict the sea level at the eastern open boundary that near the Dalian, which may be another reason for the disagreement between the observed and simulated seasonal variations of the tidal constituents

P11L21-P11L23: while the seasonal variations at Dalian were not reproduced by the test mechanisms, including seasonally varying stratification, vertical eddy viscosity and mean sea level.

**(3) Reviewer 3, point 19:**

You haven't addressed this. There needs to be an outer boundary somewhere, but it is not clear that the boundary should be the same for summer and winter. You could, for example, have calculated summer and winter boundaries using seasonal altimetry, for a 5th & 6th experiment. This is one of the other possible mechanisms for the observed result at Dalian and should be mentioned in the discussion.

You might also mention atmospheric variations as a possible mechanism at Dalian, particularly as you cannot reproduce the S2 change with your experiments.

**Reply:**

Thank you for your constructive comments. As shown in the response to the first comment of Reviewer #3, estimating the seasonal cycle from satellite altimetry data is possible, but limited to a few regions where the signal is large enough to exceed the background noise level (Muller et al., 2014). As mentioned in Shum et al. (1997) and Fang et al. (2004), the harmonic parameters derived from TOPEX/Poseidon along-track altimetry in the shallow area (e.g., the Bohai Sea) are not as accurate as in deep oceans, so if the altimetry data can be used to extract the seasonal variations of tides at the eastern open boundary of the Bohai Sea is still a problem. St-Lautent et al. (2008) used a numerical model to simulate the tides from August 2003 to August 2004 and the tidal constituents at the open boundaries were held constant. The tidal forcing at the open boundaries with constant harmonic parameters was also used in the numerical experiments to investigate the seasonality of tides in several studies, such as Huess and Andersen (2001), Kang et al. (2002), Kagan and Sofina (2010) and Devlin et al. (2018). Within the period to revise the manuscript, we do not have the ability to explore the problem. We are sorry that the seasonally varying open boundary conditions are not tested in the revised manuscript. However, this possible mechanism is added in the discussions, as shown in the following 'Changes'. In addition, the mentioned atmospheric variation is also added to be the possible mechanism in the discussions of the revised manuscript. Using the satellite altimetry data to investigate the seasonality of tides is interesting and will be tried in our following study. Changes:

P10L14-P10L16: In addition, the constant harmonic parameters were used to predict the sea level at the eastern open boundary that near the Dalian, which may be another reason for the disagreement between the observed and simulated seasonal variations of the tidal constituents

P11L4-P11L5: It may be related with the atmospheric variations and should be further investigated in future studies.

(4) Reviewer 3 (points 2 and 11) raised the data selection, but I accept your argument that you have used all the publicly available data, although this limits the study. You might consider adding a line in the discussion to this effect, to assist anyone trying to improve data availability. **Reply:**

Thank you for understanding the limitation of only using sea level data at E2 and Dalian in this study. According to your suggestion, we have added the discussion of this limitation in the revised manuscript, as shown in the following 'Changes'.

**Changes:**

P10L8-P10L9: It is a limitation that only the sea level observations at E2 and Dalian are analyzed, so the studies using much more non-publicly available data to further investigate the seasonality of tides in the Bohai Sea are encouraged.

(5) Reviewer 3 (point 3) mentions the use of literature, which could be better. For example, P2 L2 you might rearrange as "There is a seasonal variability of the M2 harmonic constants in the seas adjacent to Korea (Kang et al. 1995)", and similar for neighbouring sentences. Having done this, refocusing on the science rather than the scientist, the repetition should stand out, and you might condense it. However, this is a style point, and I don't insist on it.

**Reply:**

Thank you for your constructive suggestions. In the second paragraph of the introduction, we want to say that the seasonality of  $M_2$  tide have received considerable attention in the coastal

areas and polar regions. So, we use to the expression in the manuscript to indicate that several scientists have found and investigated the seasonal variations of the  $M_2$  tide in the areas around the world. Thank you for your toleration of the style, and we did not modify them in the revised manuscript.

(6) I suggest you move the paragraph about ice (P7L10) to the previous section (P7L3), then this mechanism is dismissed before you start the experiments.

**Reply:**

Thank you for your constructive comment. According to your suggestion, we have moved the sentences about ice in the Bohai Sea to the previous section, as shown in the following 'Changes'.

**Changes:**

P7L5-P7L10: The Bohai Sea in north China freezes to varying degrees every winter for approximately 3–4 months (Su and Wang, 2012). According to the back-effect connection of the coastal shelf and open ocean via resonance mechanisms (Arbic et al., 2009; Arbic and Garrett, 2010), sea ice may be important to the seasonality of principal tidal constituents. However, Zhang et al. (2019) performed numerical experiments with a three-dimensional ice-ocean coupled model and found that the damping effect of sea ice on the astronomical tides was almost negligible in the Bohai Sea. Therefore, ice coverage was not considered in this study.

(7) P3 L20: You don't state which constituents you have fitted in SHA. This is an important point for anyone attempting to reproduce your work, especially for this study. For example if MA2, MB2 were used, they may contain most of the seasonal modulation of M2, and M2 would not change. Please at least state the number of constituents fitted in the SHA and EHA at E2 and Dalian, and your model, and preferably list them (this can be in supplementary info if necessary).

**Reply:**

Thank you for your constructive comments. In Sections 2.2, 2.3 and 2.4, the general

descriptions of CHA, SHA and EHA are shown, respectively. As the seasonal variations of the principal tidal constituents are investigated, the MA2 and MB2 are not considered to show the seasonality of M2 tide in this study. In the manuscript, the selections of the analyzed constituents in CHA, SHA and EHA are shown in the Section 3.1 for that at E2 and Section 3.2 for that at Dalian, respectively. For examples, "Therefore, when the monthly analysis was performed in SHA, the automated constituent selection algorithm in T\_TIDE was used to determine the analysed constituents; in addition, the unresolved constituents P1 and K2 were inferred from K1 and S2, respectively" (P4L20-P4L22 in the revised manuscript). "When EHA was used to directly analyse the sea level observations at E2, the harmonic parameters of M2, K1, S2 on, P1 and K2 were estimated together, among which the harmonic parameters of M2, K1, S2 and O1 were assumed to be temporally varying and those of P1 and K2 were assumed to be constant" (P4L24-P4L26 in the revised manuscript). "The results of the final 30 d were used to calculate the harmonic parameters using CHA with the automated constituent selection algorithm" (P7L26-P7L27 in the revised manuscript).

(8) P2 L30: I was curious as to what "nearby" meant so had to look it up from your grid reference in the reply to reviewer. I suggest: "As indicated in Feng et al. (2015), the stations at Dalian and Laohutan (7km away) have data ...".

**Reply:**

Thank you for your constructive comment. According to your suggestion, we have modified it in the revised manuscript, as shown in the following 'Changes'. Changes:

P2L31-P2L32: As indicated in Feng et al. (2015), the stations at Dalian and Laohutan (7 km away) have data for the period 1975-1990 and 1991-1997, respectively.

(9) P1line 13 "an enhanced...".

**Reply:**

Thank you for your constructive comment. According to your suggestion, we have modified it in the revised manuscript, as shown in the following 'Changes'.

**Changes:**

P1L11-P1L14: The seasonal variation of tides plays a significant role in water level changes in coastal regions. In this study, seasonal variations of four principal tidal constituents, M2, S2, K1, and O1, in the Bohai Sea, China, were studied by applying an enhanced harmonic analysis method to two time series: one-year sea level observations at a mooring station (named E2) located in the western Bohai Sea and 17-year sea level observations at Dalian.

(10) P3L26: Suggest: "In contrast, the harmonic parameters are assumed to be constant in CHA and constant within each month in SHA.".

**Reply:**

Thank you for your constructive comment. According to your suggestion, we have modified it in the revised manuscript, as shown in the following 'Changes'. Changes:

P3L26-P4L1: In contrast, the harmonic parameters are assumed to be constant in CHA and constant within each month in SHA.

(11) P8L4: Why 0.2m when Figure 8 has 0.4m difference summer-winter?.

**Reply:**

Thank you for your constructive comment. Yes, the difference between the maximum value and minimum value of mean sea level at Dalian is about 0.4 m, as shown in Figure 8b. We have calculated the averaged values of mean sea level in summer (June to August, as defined in the manuscript) and winter (December to February of the following year, as defined in the manuscript), and the difference is about 0.2 m at E2. Considering that the globally averaged change in traditional mean sea level is on the order of 0.001 m/yr (Chen et al., 2017; Quartly et

al., 2017), the 0.2-m increase of water depth is used in the manuscript. We are sorry that the difference between the averaged value of mean sea level in summer and that in winter at E2 is not shown in the manuscript, and we have added it in the revised manuscript, as shown in the following 'Changes'.

**Changes:**

P8L5-P8L7: As the difference between the averaged mean sea level in summer and that in winter was about 0.2 m, Exp4 included 0.2-m increase of water depth to test the influence of mean sea level.

(12) P11 line 7: check the grammar.

**Reply:**

Thank you for your comment. We are sorry that the grammar is error and the 'was' should be deleted. We have modified it in the revised manuscript.

Changes:

P11L11-P11L12: The M2 amplitude at E2 had an annual cycle, while that at Dalian showed a semi-annual cycle.

(13) p11 line 11: Do you mean it's semi-annual?

**Reply:**

Yes, we want to say that the  $S_2$ ,  $K_1$  and  $O_1$  phase lags show semi-annual cycles. Thank you for your comment. We have modified the sentences to be 'The maxima of the  $S_2$ ,  $K_1$  and  $O_1$  phase lags at E2 appeared twice a year' in the revised manuscript.

**Changes:**

P11L16-P11L17: The maxima of the  $S_2$ ,  $K_1$  and  $O_1$  phase lags at E2 appeared twice a year, which was the same as those of  $K_1$  and  $O_1$  and different from that of  $S_2$  at Dalian.

[revised manuscript text omitted]

a Vertical eddy viscosity coefficient.

Figure 1. (a) General location of the Bohai Sea (rectangle with dashed lines); and (b) locations of the observation stations (red stars), E2 and Dalian, in the Bohai Sea, and bathymetry of the Bohai Sea (colours).